# Transcriptional control of motor pool formation and motor circuit connectivity by the LIM-HD protein Isl2

**Yunjeong Lee[1], In Seo Yeo[1], Namhee Kim[2], Dong-Keun Lee[1], Kyung-Tai Kim[3], Jiyoung Yoon[1], Jawoon Yi[1], Young Bin Hong[4,5], Byung-Ok Choi[4], Yoichi Kosodo[6], Daesoo Kim[7], Jihwan Park[1], Mi-Ryoung Song[1]\***

[1]School of Life Sciences, Gwangju Institute of Science and Technology, Oryong-dong, Buk-gu, Gwangju, Republic of Korea; [2]Fermentation Regulation Technology Research Group, World Institute of Kimchi, Gwangju, Republic of Korea; [3]Jeonbuk Department of Inhalation Research, Korea Institute of Toxicology, Jeongeup-si, Republic of Korea; [4]Department of Neurology, Samsung Medical Center, Sungkyunkwan University School of Medicine, Seoul, Republic of Korea; [5]Department of Biochemistry, College of Medicine, Dong-A University, Busan, Republic of Korea; [6]Korea Brain Research Institute, Daegu, Republic of Korea; [7]Department of Brain and Cognitive Sciences, Korea Advanced Institute of Science and Technology (KAIST), Daejeon, Republic of Korea

**\*For correspondence:**
msong@gist.ac.kr

**Competing interest:** The authors declare that no competing interests exist.

**Abstract** The fidelity of motor control requires the precise positional arrangement of motor pools and the establishment of synaptic connections between them. During neural development in the spinal cord, motor nerves project to specific target muscles and receive proprioceptive input from these muscles via the sensorimotor circuit. LIM-homeodomain transcription factors are known to play a crucial role in successively restricting specific motor neuronal fates. However, their exact contribution to limb-based motor pools and locomotor circuits has not been fully understood. To address this, we conducted an investigation into the role of Isl2, a LIM-homeodomain transcription factor, in motor pool organization. We found that deletion of *Isl2* led to the dispersion of motor pools, primarily affecting the median motor column (MMC) and lateral motor column (LMC) populations. Additionally, hindlimb motor pools lacked Etv4 expression, and we observed reduced terminal axon branching and disorganized neuromuscular junctions in *Isl2*-deficient mice. Furthermore, we performed transcriptomic analysis on the spinal cords of *Isl2*-deficient mice and identified a variety of downregulated genes associated with motor neuron (MN) differentiation, axon development, and synapse organization in hindlimb motor pools. As a consequence of these disruptions, sensorimotor connectivity and hindlimb locomotion were impaired in *Isl2*-deficient mice. Taken together, our findings highlight the critical role of *Isl2* in organizing motor pool position and sensorimotor circuits in hindlimb motor pools. This research provides valuable insights into the molecular mechanisms governing motor control and its potential implications for understanding motor-related disorders in humans.

## Editor's evaluation

This paper will be of interest to developmental biologists who study the gene regulatory mechanisms necessary for neuronal identity and circuit assembly. The study presents important findings regarding the role of the LIM homeodomain transcription factor Isl2 in the development of spinal motor neurons. While the importance of Isl2 for the acquisition of axial and visceral motor neuron

development was already described in the literature, the data convincingly describe an additional role in the differentiation of a subset of limb-innervating motor neurons.

## Introduction

The establishment of motor circuits in the spinal cord involves complex processes including the organization of motor neurons (MNs) into specific groups known as motor columns and motor pools along the rostrocaudal axis (*Romanes, 1964*; *Vanderhorst and Holstege, 1997*). In each columnar group, MNs are further segregated into distinct motor pools. The development of this diverse repertoire of motor pools involves the sequentially activated various genetic programs during spinal cord development. Combinatorial gene expression of LIM-homeodomain (HD) transcription factors determines MN identity and positional information across multiple segments, including the median motor column (MMC), the hypaxial motor column (HMC), the lateral motor column (LMC), and the preganglionic column (PGC) (*Pfaff et al., 1996*; *Tsuchida et al., 1994*). *Hox* genes, crucial for body patterning, also contribute to segmental identity by selectively expressing in postmitotic MNs. The absence or misexpression of *Hox* genes such as *Hox6* and *Hox10* can lead to disorganization and misdirected targeting of LMC MNs toward specific limb muscles (*de la Cruz et al., 1999*; *Wu et al., 2008*). Foxp1, which interacts with *Hox* genes, is also essential for determining MN fates in the LMC, whose absence leads to scattered and misspecified LMC neurons as demonstrated in *Foxp1* mutant mice (*Dasen et al., 2008*). Nevertheless, despite the involvement of *Hox* networks, the intricate three-dimensional organization of individual motor pools may require additional regulatory mechanisms, as elongated motor pools can span several spinal cord segments.

More than 50 limb muscles in tetrapods are innervated by distinct limb motor pools, suggesting the involvement of diverse molecular mechanisms in shaping individual limb motor pools during spinal cord development (*Sullivan, 1962*). Specific LMC motor pools express ETS factors Etv4 and Etv1, playing a role in acquiring motor pool identities. In *Etv4* mutant mice, the position and terminal arborization of specific motor pools were perturbed, resulting in disorganized motor pools and impaired motor control (*Livet et al., 2002*; *Vrieseling and Arber, 2006*). Gdnf is suggested to act as a peripheral signal derived from limb tissues, guiding major axon bundles toward the hindlimb (*Gould et al., 2008*; *Kramer et al., 2006*). Furthermore, Gdnf signaling appears to induce Etv4 expression in certain motor pools, as deletion of *Gdnf,* its receptor *Ret*, or *Gfra1* results in extinguished *Etv4* expression and affected terminal arborization of motor axons in target muscles (*Haase et al., 2002*; *Livet et al., 2002*). However, it remains unclear whether the development of Etv4-expressing motor pools solely depends on Gdnf signaling.

Insulin-related, LIM-homeodomain protein 1 (Isl1) and insulin-related, LIM-homeodomain protein 2 (Isl2) are two closely related LIM-HD transcription factors, sharing approximately 75% of their identity, and both are expressed in postmitotic MNs with overlapping patterns. Isl1 is first detected in all newborn MNs shortly after these neurons exit the mitotic cycle at E9.5 (*Pfaff et al., 1996*; *Thaler et al., 2004*), while *Isl2* begins its expression slightly later, at E10.5 (*Thaler et al., 2004*). Genetic studies in mice and zebrafish reveal that *Isl1* plays a significant role, as its removal induces defects in motor neuronal fates, axonal navigation, neurotransmitter identity, and electrical excitability (*Liang et al., 2011*; *Moreno and Ribera, 2014*; *Pfaff et al., 1996*; *Wolfram et al., 2012*). Although Isl1 has been considered more crucial than Isl2, evidence suggests that *Isl2* is also essential in MN development. In *Isl2*-null mice, thoracic MNs are affected, leading to scattered or misspecified MN subsets (*Thaler et al., 2004*). Similarly, misexpression of the dominant negative form of *Isl2* or knockdown (KD) of *Isl2* in zebrafish results in mispositioned MN cell bodies and defective axon growth (*Hutchinson and Eisen, 2006*; *Segawa et al., 2001*). Moreover, successful rescue experiments using *isl2* mRNA in *isl1* morphants demonstrate that both Isl1 and Isl2 are equally potent in neurons (*Hutchinson and Eisen, 2006*). While *Isl2* initially coexists with *Isl1* in pan-MNs, it later becomes enriched in lateral LMC (LMCl), which lacks Isl1 expression (*Thaler et al., 2004*). This suggests a possible independent role for *Isl2* in LMCl beyond its reported function in thoracic MN subsets. Recent advances in transcriptomic analysis reveal detailed spatiotemporal changes in the gene expression of Islet transcription factor during MN development, indicating persistent expression of Isl2 in postnatal MNs (*Amin et al., 2021*; *Catela et al., 2022*; *Delile et al., 2019*; *Rayon et al., 2021*). Overall, these findings suggest that the

role of Isl2 in motor neuronal development may have been underestimated and warrants further investigation, despite the considerable similarities or redundancies in Islet transcription factor.

In this study, we have demonstrated the crucial role of Isl2 in the development and differentiation of hindlimb motor pools. Isl2 regulates the position of motor neuronal cell bodies and directs the assembly of the motor neuronal circuit. Our findings indicate that Isl2 is enriched in proximal motor pools expressing Etv4, and it is involved in neuromuscular junction (NMJ) formation, axonal and dendritic arborization, and hindlimb movement. Through transcriptomic analysis, we have identified *Isl2* as a master regulator of various genes involved in the development and differentiation of hindlimb motor pools. Our results indicate that LIM-HD transcription factors, together with Hox and ETS transcription factors, fine-tune the organization of hindlimb motor pools.

## Results

### Spatiotemporal expression of Islet transcription factors during MN development

Islet transcription factors *Isl1* and *Isl2* are expressed immediately after MN generation. However, the detailed expression of these factors during MN development at the cellular level remains uncertain. To address this, we re-analyzed published single-cell RNA-sequencing data obtained from E12 MNs, sorted from the cervical to lumbar regions of the spinal cord, focusing on changes in their expression levels (*Amin et al., 2021*). We categorized approximately 3483 developing MNs into eight subgroups representing major motor columns, such as pMN, immature, MMC, phrenic/hypaxial motor column (P/HMC), LMCm, LMCl, PGCa, and PGCb (*Figure 1a and c*). Uniform Manifold Approximation and Projection (UMAP) analysis revealed the progression of pMN progenitors into immature MNs, MMC, P/HMC, LMCm, LMCl, PGCa, and PGCb MNs. In earlier immature MNs, we observed high *Isl1* expression, which slightly decreased in most differentiated MNs, such as MMC, P/HMC, LMCm, and PGCa MNs, but not LMCl MNs, which did not express *Isl1*. *Isl2* expression emerged later than *Isl1* in immature MNs, persisting in MMC and P/HMC neurons. Most LMC neurons showed low but even expression of *Isl2*, while a few PGCa and b cells expressed *Isl2* (*Figure 1b*). At E10.5, during MN generation, Olig2$^+$ pMNs were initially found in the middle part of the ventral spinal cord, co-expressing Lhx3, a marker for pMNs and MMCs (*Lee et al., 2005*; *Thaler et al., 2002*; *Figure 1d*). As these cells migrated, they transiently expressed markers for postmitotic MNs Isl1 and Hb9. *Isl2* expression emerged slightly later, found in more lateral regions across all levels analyzed. To analyze the distribution of Isl2 during the MN maturation, we compared the relative position of three populations: Olig2$^+$ pMNs, Olig2$^+$Isl1$^+$ immature MNs, and *Isl2*-expressing MNs. Contour and density assessments revealed that *Isl2* expression began slightly after Isl1 expression in immature MNs and persisted in postmitotic MNs along with Isl1 (*Figure 1d*). At E12.5, when motor columns became distinct, we identified them based on the expression of motor neuronal markers, including Foxp1 (marker for LMC neurons), Isl1 (marker for LMCm and PGC neurons), Lhx3 (marker for MMC neurons), nNOS (marker for PGC neurons), and Etv4 (marker for some LMC motor pools). Motor pools expressing Etv4 were located in the LMC neurons but not at the thoracic level (*Figure 1e*). Isl2 exhibited broad expression across all motor columns, unlike other LIM-HD transcription factors that showed specific expression patterns. Overall, while *Isl2* is present in differentiated MNs, it is relatively enriched in MMC and LMCl populations, where its expression is more prominent compared to Isl1.

### Settling position of MNs is impaired in the absence of *Isl2*

Given the broad expression of *Isl2* in postmitotic MNs, we examined the organization of motor columns in *Isl2*-null mice (*Thaler et al., 2004*). The pan-motor neuronal expression of *Isl2* across the multiple segments of the spinal cord suggests that additional motor neuronal populations could also be affected by the absence of *Isl2*. To explore unknown phenotypes, we conducted further investigations using the mutant mouse line previously generated and analyzed by *Thaler et al., 2004*. At E13.5, the number of Hb9$^+$Lhx3$^+$ MMC and Foxp1$^+$ LMC neurons remained unchanged at the brachial and lumbar levels (*Figure 2a–c*). At the thoracic level, the number of MMC and HMC neurons remained unaffected, but we observed a decrease in the number of Isl1$^+$nNOS$^+$ PGC neurons (*Figure 2b and c* and *Supplementary file 2*). *Isl2*-null mice also showed significantly reduced expression of nNOS and pSMAD in PGC neurons, consistent with previous studies (*Dasen et al., 2008*; *Harrison et al., 1999*;

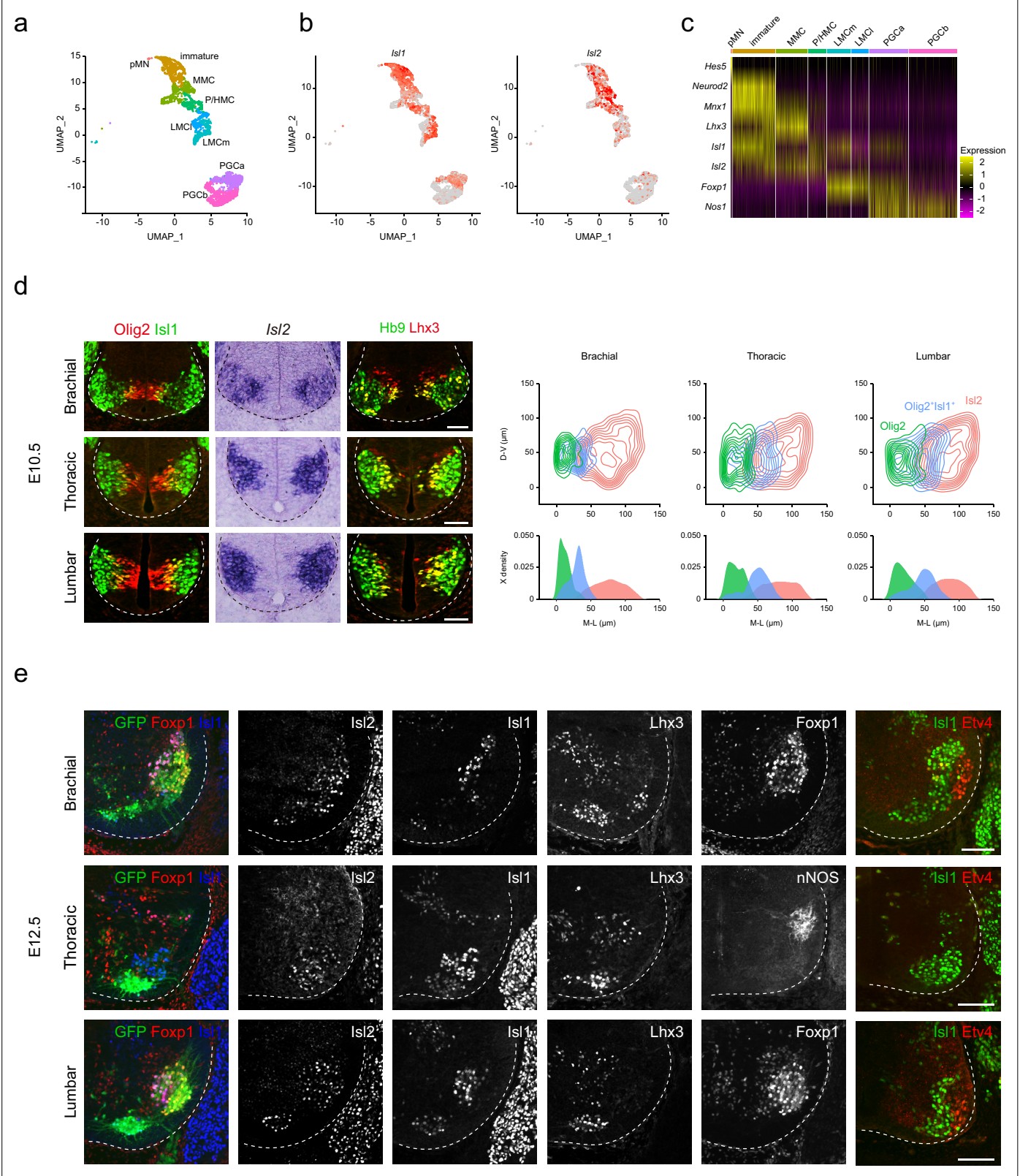

**Figure 1.** Spatiotemporal dynamics of *Isl2* expression and heterogeneity of motor neurons (MNs). (**a**) Uniform Manifold Approximation and Projection (UMAP) visualization of scRNA-seq data based on et al.'s study, highlighting distinct MN lineages: pMN progenitors, immature MNs, MMC, P/HMC, LMCm, LMCl, PGCa, and PGCb neurons, each depicted in distinct clusters. (**b**) UMAP representation showing the differential expression of *Isl1* and *Isl2*, across MN clusters. (**c**) Heatmap demonstrating the dynamic gene expression profiles of *Isl1*, *Isl2*, and other marker genes within MN clusters.

*Figure 1 continued on next page*

*Figure 1 continued*

(**d**) Expression of Olig2, Isl1, *Isl2*, Hb9, and Lhx3 in E10.5 spinal cords. Contour plots and medio-lateral density plots highlighting Olig2+ MN progenitors (green), Olig2+Isl1+ newborn MNs (blue), *Isl2*+ MNs (pink) across all spinal cord levels. Scale bars, 50 µm. (**e**) Expression of Isl1, Isl2, Lhx3, Foxp1, nNOS, Etv4, and Hb9::GFP in E12.5 spinal cords. Scale bars, 100 µm.

*Figure 2b*). Additionally, the overall distribution of MNs was altered in *Isl2*-null mice (*Figure 2a and b*). At the limb level, MN cell bodies were slightly scattered, located between the MMC and LMC neurons. At the thoracic level, ectopic cells were dispersed dorsally to the MMC neurons. Ectopic cells at the lumbar level mainly belonged to LMCl MNs, expressing Lhx1+ (58%), Foxp1+ (53%), and Lhx3+

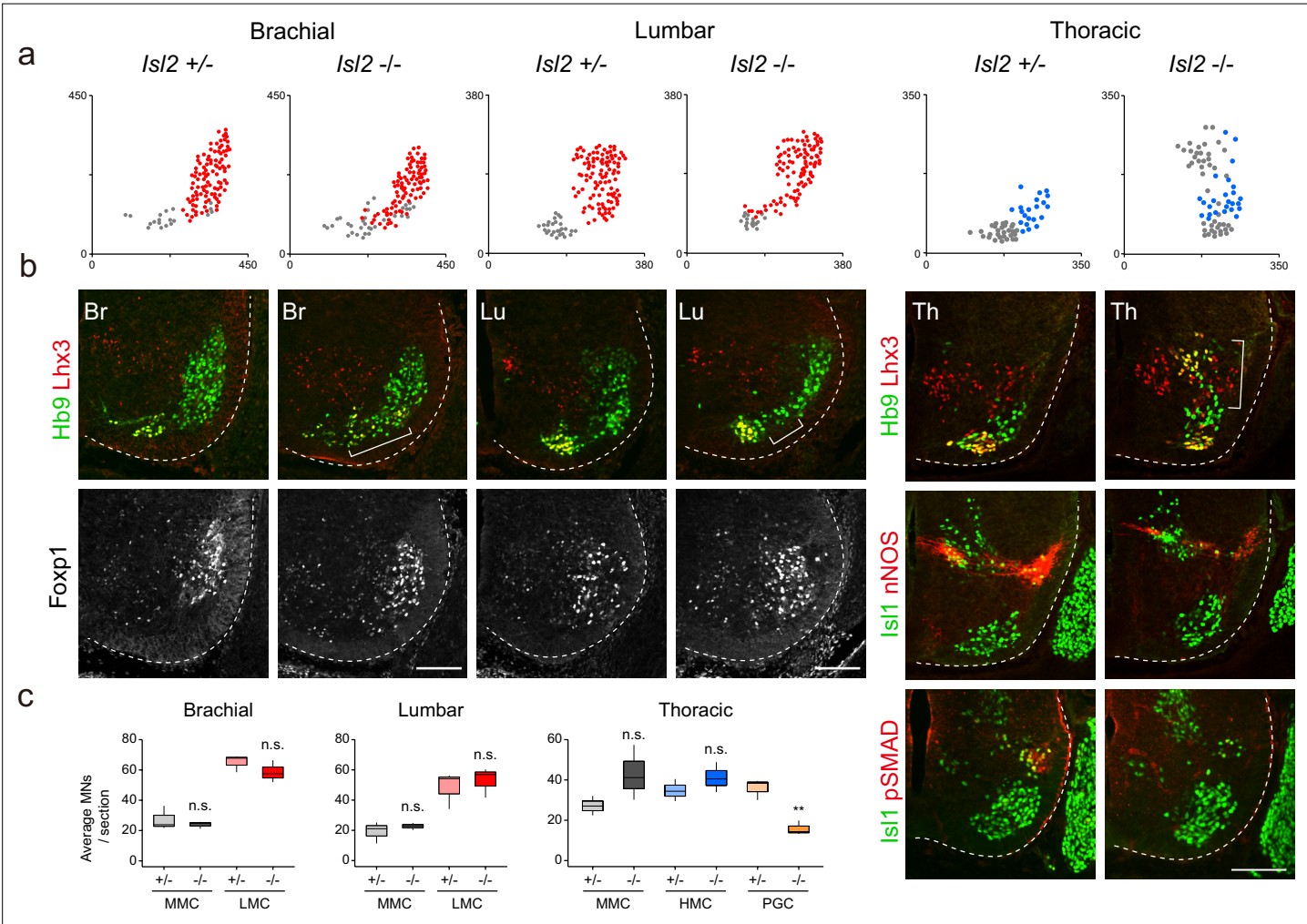

**Figure 2.** Disrupted motor column organization in *Isl2*-null mice. (**a**) Spatial representation of motor neuron (MN) subtypes: median motor column (MMC) (gray, Lhx3+Hb9+), lateral motor column (LMC) (red, Foxp1+), and preganglionic column (PGC) (blue, Isl1+ nNOS+) neurons from the images in b. X and y coordinates denote the medial-to-lateral and ventral-to-dorsal axes, respectively, in µm. (**b**) Expression of Hb9, Lhx3, Foxp1, Isl1, nNOS, and pSMAD in E13.5 mouse embryonic spinal cords. Brackets mark ectopic MNs observed in *Isl2*-null mice. (**c**) Quantification of MMC (gray), LMC (red), hypaxial motor column (HMC) (blue), and PGC (yellow) neurons across brachial (C6–C8), lumbar (L2–L4), and thoracic spinal segments. Average MN number per mouse. Box plots illustrate distribution with median (center line), first and third quartiles (box boundaries), and 10th–90th percentiles (whiskers). Data from three mice per group, analyzed using unpaired Student's t-test; **p<0.01; n.s. indicates not significant. Refer to *Supplementary file 2* and source data for detailed statistics. Scale bars, 100 µm.

The online version of this article includes the following source data and figure supplement(s) for figure 2:

**Source data 1.** Quantification of motor neuron (MN) subtypes at E13.5 in *Isl2* +/- and *Isl2*-null mice, categorized by brachial, lumbar, and thoracic levels.

**Figure supplement 1.** Analysis of ectopic motor neurons (MNs) in *Isl2* knockout (KO) mice.

**Figure supplement 1—source data 1.** Quantification of ectopic motor neurons (MNs) at E13.5 in the lumbar spinal cords in *Isl2*-null mice.

MMC MNs (32%) (*Figure 2—figure supplement 1*). Our findings align with previous results, including a reduced number of visceral MNs, reduced nNOS expression, and abnormal dorsomedial migration of MNs, reported by *Thaler et al., 2004* Our study additionally reveals the appearance of ectopic MNs at the limb levels, primarily consisting of MMC and LMCl cells.

To further characterize the scattered MNs at the limb level, we examined the organization of motor pools in *Isl2*null (knockout [KO]) and *Isl2* conditional KO (cKO) mice at the brachial level. *Isl2* cKO mice were generated by crossing *Isl2^{F/F}* mice with *Isl2 +/-; Olig2^{Cre}* mice (*Figure 3—figure supplement 1a and b*; *Dessaud et al., 2007*; *Kong et al., 2015*). In situ hybridization analysis showed specific downregulation of *Isl2* transcript levels among MNs (*Figure 3—figure supplement 1c*). We assessed MNs based on the expression of marker genes for motor columns and pools, such as Hb9, Lhx3, Foxp1, Isl1, Lhx1 (marker for LMCl neurons), Etv4, and Scip (marker for MMC/HMC MNs and some motor pools), in consecutive transverse spinal cord sections (*Figure 3a*). At the thoracic level, the previous study reported that the organization of the MMC is disrupted in the absence of *Isl2* (*Thaler et al., 2004*). Similarly, we found that MMC neurons were scattered dorso-laterally and located ectopically within LMC MNs or immediately dorsal to motor columns in *Isl2* KO and cKO brachial spinal cords (see *Figure 2*, *Figure 3a and b*, *Figure 3—figure supplements 2 and 3*). The distribution of LMCm was relatively spared, whereas LMCl neurons spread more medially when determined by Foxp1, Isl1, and Lhx1 immunoreactivity. The distribution of major brachial motor pools, defined by Etv4 and Scip expression, remained unchanged in *Isl2* mutant mice. At the C6 and C8 levels, the position of major motor pools such as cutaneous maximus (CM), latissimus dorsi (LD), and flexor carpi ulnaris (FCU) motor pools was unaffected in *Isl2* KO and cKO mice (*Figure 3a*, *Figure 3—figure supplements 2 and 3*; *Dasen et al., 2005*; *Livet et al., 2002*). Consequently, the number of individual motor pools also remained unchanged in *Isl2* mutant mice (*Figure 3c*). Thus, at the brachial level, MMC and LMCl neurons were mostly scattered, while the position of other populations remained relatively normal.

Next, we reconstructed the detailed motor pool distribution at L2 and L4 levels by analyzing the expression of motor neuronal markers, Hb9, Lhx3, Foxp1, Scip (marker for MMC/HMC neurons), Nkx6.1 (marker for MMC and LMCm neurons), Lhx1, Isl1, and Etv4 (*Figure 4*). Scip^+ MMC/HMC neurons were scattered dorsally, and Nkx6.1^+ LMCm and Lhx1^+ LMCl neurons were scattered medially in both *Isl2* KO and cKO mice, as shown in contour, density, or spatial plots (*Figure 4a and b*, *Figure 4—figure supplements 1 and 2*). Lhx1^+ LMCl neurons were scattered medially, while Nkx6.1^+ LMCm neurons were less distinct in *Isl2* mutant mice (*Figure 4a, b, and c* and *Figure 4—figure supplements 1 and 2*). The clustering of tibialis anterior (Ta) motor pools was relatively normal in *Isl2* mutant mice (*Figure 4c*). Remarkably, the number of Etv4-expressing cells, including gluteus (Gl), rectus femoris (Rf), and tensor fasciae latae (Tfl) motor pools in the L2 and L4 segments, was significantly downregulated, as shown in contour and spatial plots (*Figure 4a, c, and d* and *Figure 4—figure supplements 1 and 2*). Taken together, MMC neurons at all levels and some LMC motor pools at the lumbar level were dispersed in the absence of *Isl2*.

## Isl2 induces *Etv4* transcripts in lumbar motor pools

The presence of scattered MNs and the absence of Etv4 expression in *Isl2* mutant lumbar motor pools led us to hypothesize that Isl2 activates *Etv4* expression, as its absence also resulted in positional defects in MNs (*Livet et al., 2002*). To investigate this, we examined Etv4 expression in *Isl2* mutant mice during the segregation of limb motor pools from E11.5 to E13.5. Immunohistochemistry and RT-PCR analysis using spinal cord tissues revealed that both the proteins and transcripts of *Etv4* were selectively downregulated in lumbar motor pools, but not in brachial motor pools, of *Isl2* KO and *Isl2* cKO spinal cords (*Figure 5a and b* and *Figure 5—figure supplement 1*).

To further test whether acute downregulation of *Isl2* is sufficient to abolish *Etv4* transcription, we knocked down *ISL2* expression in the chick neural tube using siRNA. Efficient KD of *ISL2* was confirmed in a cell line misexpressing HA-tagged ckISL2 (*Figure 5—figure supplement 2*). When *ISL2* siRNA was introduced into the chick neural tube by in ovo electroporation, the number of ISL2-expressing cells and the intensity of ISL2 in these cells were reduced, while the total number of MNs labeled with MNR2, a marker expressed among MNs and progenitors in chicken embryos, remained unchanged (*Tanabe et al., 1998*; *Figure 5c*). Consistent with the results in mice, the *ETV4* transcript level was significantly downregulated in chick *ISL2*-KD MNs (*Figure 5c*). Collectively, these results

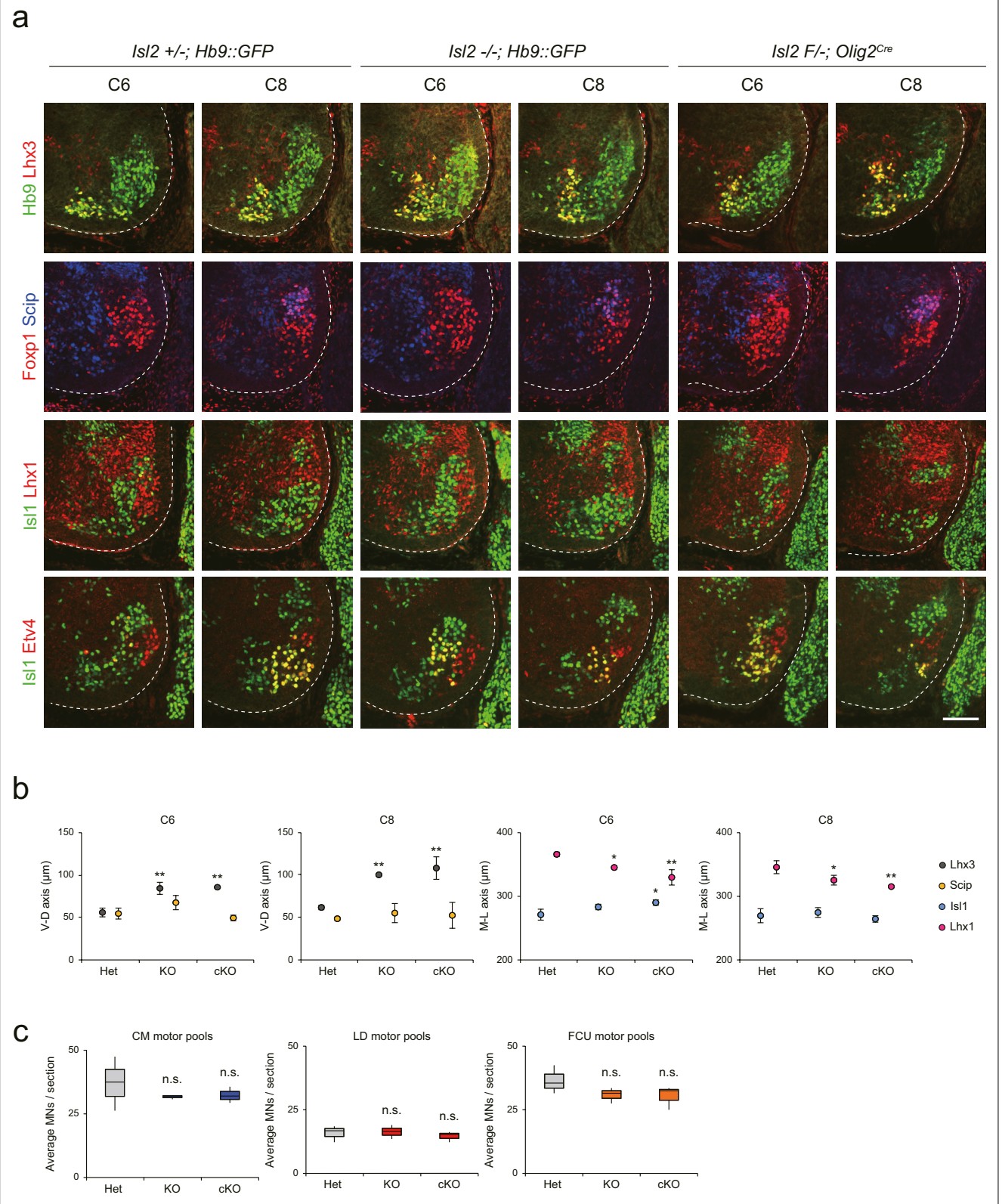

**Figure 3.** Distribution of lateral motor column (LMC) motor pools in brachial spinal cords of control and *Isl2* mutant mice. (**a**) Immunofluorescence images of motor neurons (MNs) labeled with Hb9, Lhx3, Foxp1, Scip, Isl1, Lhx1, and Etv4 in adjacent sections of C6 and C8 spinal cords from Het (+/-), *Isl2* knockout (KO) (-/-), and *Isl2* conditional knockout (cKO) (*Isl2 F/-; Olig2^Cre^*) mice. (**b**) Average ventro-dorsal distribution of MMC (Hb9$^+$Lhx3$^+$) and MMC/HMC (Hb9$^+$Foxp1$^-$Scip$^+$) neurons, and medio-lateral distribution of LMCm (Hb9$^{low}$Isl1$^+$) and LMCl (Hb9$^{high}$Lhx1$^+$) neurons in embryos with each

*Figure 3 continued on next page*

*Figure 3 continued*

genotype. n=3 mice for each genotype; standard deviation (SD) is shown; one-way ANOVA with Bonferroni's post hoc test; Het vs. KO or cKO, **p<0.01, *p<0.05. (**c**) Average cell count per embryo in flexor carpi ulnaris (FCU) (Foxp1+Scip+), cutaneous maximus (CM) (Isl1+Etv4+), and latissimus dorsi (LD) (Isl1−Etv4+) motor pools. Box plots illustrate distribution with median (center line), first and third quartiles (box boundaries), and 10th-90th percentiles (whiskers). One-way ANOVA with Bonferroni's post hoc test; n.s. indicates not significant. See *Supplementary file 2* and source data for detailed n and statistics. Scale bar, 100 μm.

The online version of this article includes the following source data and figure supplement(s) for figure 3:

**Source data 1.** Average ventro-dorsal distribution of MMC (Hb9+Lhx3+) and MMC/HMC (Hb9+Foxp1−Scip+) neurons as well as medio-lateral distribution of LMCm (Hb9lowIsl1+) and LMCl (Hb9highLhx1+) neurons (*Figure 3b*).

**Figure supplement 1.** Generation of novel *Isl2* flox mouse line.

**Figure supplement 1—source data 1.** An unedited raw agarose gel picture and a labeled gel picture of genotyping PCR for *Figure 3—figure supplement 1b*.

**Figure supplement 2.** Contour plots for motor neuron (MN) subtypes and motor pools in the brachial spinal cords.

**Figure supplement 3.** Spatial plots depict position of individual neurons comprising motor columns and motor pools indicated from three different animals shown in *Figure 3*.

suggest that Isl2 is necessary for gene transcription of *Etv4* in MNs in a cell-autonomous manner, which may correlate with the correct positioning of motor pools.

## Transcriptomic analysis to define genes is associated with the development of lumbar motor pools

Due to the more severe effect observed in the absence of *Isl2* on lumbar motor pools at the lumbar level compared to other axial levels, we sought to identify sets of genes under the control of *Isl2*, particularly in the lumbar segment. We applied two search criteria. First, we focused on downregulated genes, as Isl2 is expected to act as an activator similar to Isl1 (*Lee et al., 2008*; *Lee and Pfaff, 2003*). Second, we compared differentially expressed genes (DEGs) at the brachial and lumbar levels to select genes specifically downregulated at the lumbar level. To achieve this, we conducted bulk RNA-seq analysis using E12.5 brachial and lumbar ventral spinal cords. To obtain MN-enriched tissues, we dissected the ventral spinal cord from an open-book preparation. Segmental identity of the spinal cord was verified according to the expression of the correct *Hox* code at each axial level (*Hayashi et al., 2018*). The DEG analysis between *Isl2* KO and *Isl2* heterozygote mice revealed 140 genes downregulated at the brachial levels and 159 genes downregulated at the lumbar levels ($<-0.5$ log$_2$-FC, raw p<0.05). We then identified DEGs that overlapped with the list of genes from the scRNA-seq dataset obtained from embryonic MNs to further identify MN-specific genes (*Amin et al., 2021*). As a result, 44 and 87 DEGs were specifically downregulated at the brachial and lumbar levels of *Isl2* KO spinal cords, respectively, whereas nine DEGs were commonly downregulated in both segments (*Figure 6a–e* and *Supplementary file 3a-b*). Notably, gene ontology (GO) analysis revealed that the downregulated genes at the lumbar level in *Isl2*-null embryos were predominantly associated with the neuropeptide signaling pathway, synaptic signaling, locomotion, neuron differentiation, and axon development (*Figure 6g*). In contrast, at the brachial level, no categories involving synapse organization, axon development, and neuron differentiation were detected (*Figure 6f*). To validate some of these findings, we investigated the top 20 DEGs (p-value <0.001) and confirmed their expression in *Isl2* mutant spinal cords (*Figure 6h*). For instance, *A730046J19Rik* transcript was detected in Rf motor pools at the L2 level of E12.5 spinal cords and was downregulated in *Isl2* KO. Expression of *Anxa2*, *Kcnab1*, Etv4, *Prph*, and *C1ql3* was also found in Gl motor pools, the most ventrolaterally located motor pool subsets, at the L4 level and was markedly downregulated in the *Isl2* KO spinal cords, consistent with normalized read count values.

To pinpoint the specific hindlimb motor pools affected by the downregulation of Isl2, we further analyzed the LMC motor pools using a published single-cell RNA-sequencing dataset from E12.5 mouse spinal cords (*Amin et al., 2021*). We isolated the LMC subclusters in the rostral lumbar spinal cord (L1 to L4) that expressed high levels of *Hoxd9*, *Hoxc10*, *Hoxd10*, *Hoxa11*, *Hoxc11*, and *Hoxd11* (*Figure 6—figure supplement 1*). UMAP analysis of 273 cells identified five clades: two LMCl and three LMCm subclusters (*Figure 6—figure supplement 1a–h*). Subcluster LMCl.1 (sb.LMCl.1) expressed *Isl2* and *Lhx1* but not *Etv4* and was further divided into two subclusters: sb.LMCl.1.v, consisting of

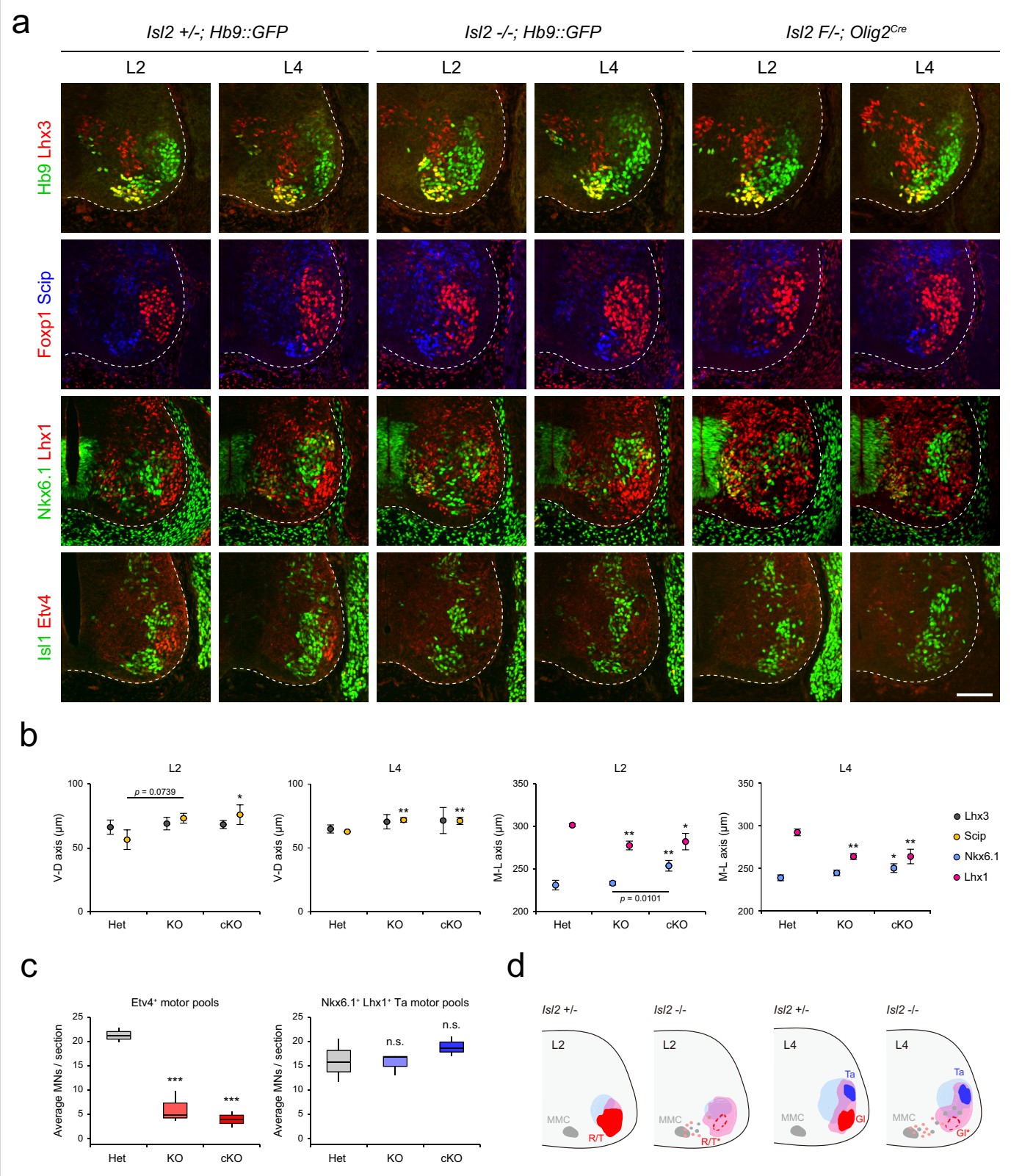

**Figure 4.** Altered lumbar motor column position in *Isl2* mutant mice. (**a**) Immunofluorescence images of motor neurons (MNs) labeled with Hb9, Lhx3, Foxp1, Scip, Nkx6.1, Lhx1, Isl1, and Etv4 in adjacent sections of L2 and L4 spinal cords from Het (+/-), *Isl2* knockout (KO) (-/-), and *Isl2* conditional KO (cKO) (*Isl2 F/-; Olig2$^{Cre}$*) mice. (**b**) Average ventro-dorsal distribution of MMC (Hb9$^+$Lhx3$^+$) and MMC/HMC (Hb9$^+$Foxp1$^-$Scip$^+$) neurons and medio-lateral distribution of LMCm (Hb9$^{low}$Nkx6.1$^+$) and LMCl (Hb9$^{high}$Lhx1$^+$) neurons in heterozygote, *Isl2* KO, and *Isl2* cKO embryos. n=3 mice for each genotype; SD

*Figure 4 continued on next page*

*Figure 4 continued*

is shown; one-way ANOVA with Bonferroni's post hoc test; **p<0.01, *p<0.05. See *Supplementary file 2* and source data for detailed n and statistics. (**c**) The average cell count per embryo in Etv4⁺ rectus femoris (Rf)/tensor fasciae latae (Tfl)/gluteus (Gl) and Nkx6.1⁺Lhx1⁺ Ta motor pools. n=3 mice for each genotype; SEM is shown; one-way ANOVA with Bonferroni's post hoc test, ***p<0.001, n.s.=not significant. Box plots illustrate data distribution with median (center line), first and third quartiles (box boundaries), and 10th-90th percentiles (whiskers). See *Supplementary file 2* and source data for detailed n and statistics. (**d**) Summary diagram depicting the position of major lumbar motor pools: rectus femoris (**R**), tensor fascia latae (**T**), tibialis anterior (**Ta**), and Gl. Misspecified motor pools are marked with asterisks. Scale bar, 100 μm.

The online version of this article includes the following source data and figure supplement(s) for figure 4:

**Source data 1.** Average ventro-dorsal distribution of MMC (Hb9⁺Lhx3⁺) and MMC/HMC (Hb9⁺Foxp1⁻Scip⁺) neurons, as well as the medio-lateral distribution of LMCm (Hb9^low^Nkx6.1⁺) and LMCl (Hb9^high^Lhx1⁺) neurons (*Figure 4b*).

**Figure supplement 1.** Motor neuron (MN) subtypes and motor pool distribution in lumbar spinal cords.

**Figure supplement 2.** Spatial plots depict position of individual neurons comprising motor columns and motor pools indicated from three different animals shown in *Figure 4*.

vasti (V) motor pools expressing *Isl2*, *Lhx1*, and *Etv1*; and sb.LMCl.1.ta, consisting of Ta motor pools expressing *Isl2*, *Lhx1*, and *Nkx6-1*. sb.LMCl.2 contained Rf, Tfl, and Gl motor pools that expressed *Lhx1*, *Nkx6-2*, and *Etv4*. sb.LMCm.1 was characterized by the expression of *Isl2*, *Isl1*, and *Etv1*, with molecular profiles similar to the adductor/gracilis (A/G) motor pools. Subclusters sb.LMCm.2 and sb.LMCm.3 were both defined by the expression of *Isl2*, *Isl1*, and *Nkx6-1* and contained hamstring (H) and gastrocnemius (Gs) motor pools, with sb.LMCm.2 (*Hoxd9*^high^ and *Aldh1a2*^high^) occupying a more rostral position than sb.LMCm.3 (*Hoxd9*^low^ and *Aldh1a2*^low^) (*Figure 6—figure supplement 1*). Thus, we successfully defined five subclusters of LMC motor pools. Among the LMC subclusters, *Isl2* expression was higher in sb.LMCl.2 than in other subclusters, suggesting a close genetic relationship between *Isl2* and *Etv4* (*Figure 6—figure supplement 1b-d*). Moreover, some genes were relatively enriched in specific motor pools, which is potentially useful for distinguishing individual motor pools. For instance, *Epha3*, *Fgf10*, and *Sema5a* transcripts were enriched in Gl motor pools and their expression was reduced in *Isl2* KO, as validated by in situ hybridization analysis (*Figure 6i*, *Supplementary file 3c*). In conclusion, our integrated approach, combining bulk and scRNA-seq analyses datasets, has proven valuable in identifying new potential motor pool markers and uncovering novel transcription target genes.

## Defective arborization and sensorimotor connectivity of *Isl2*-deficient hindlimb motor pools

The downregulation of *Etv4* and the transcriptomic analysis of *Isl2*-deficient mice suggested that these mice have similar phenotypes to those found in *Etv4* mutant mice, such as scattered cell bodies, reduced dendrite patterning, and sensorimotor connectivity (*Vrieseling and Arber, 2006*). At the lumbar level of the spinal cord, there were three motor pools that expressed Etv4: Gl, Rf, and Tfl motor pools (*Arber et al., 2000*; *De Marco Garcia and Jessell, 2008*). Thus, we sought to investigate whether Isl2 plays a crucial role in dendrite formation of these motor pools. A previous study by *Thaler et al., 2004*, had established that *Isl2* KO mice were unable to survive beyond a day after birth (*Thaler et al., 2004*). Intriguingly, our study revealed that *Isl2* cKO mice also died on the first day after birth, while 11% of *Isl2* KO mice managed to survive (*Supplementary file 1*, *Video 1*). While the precise underlying cause for this disparity in survival rates among *Isl2* KO mutant mouse lines requires further investigation, we were able to examine adult MNs in the absence of Isl2. Overall, the total number of ChAT⁺ MNs remained unchanged, and mislocalized MNs were observed in the P21 *Isl2*-null lumbar spinal cord (*Figure 7a–c*). We next examined the dendrite formation of motor pools by injecting rhodamine-dextran (Rh-Dex) retrograde tracer into individual muscles of *Isl2* KO postnatal pups at P4. In the wild-type animals, the Gl motor pools exhibited a stereotyped elongated crescent shape of the dextran-labeled dendritic arbor at the ventrolateral position in L3-L4 spinal cords (*Figure 7d*). However, in *Isl2* KO mice, dextran-labeled Gl motor pools split into two groups: 61% of cells were located at the correct ventrolateral position, and 39% were located at the ectopic ventromedial position. Regardless of position, *Isl2*-deficient Gl motor pools showed shorter and more randomly oriented dendritic arbors (*Figure 7d*). Similarly, Tfl motor pools, whose axons diverged from one of the major branches of Gl ones, also displayed shorter and lesser dendritic arbors in *Isl2* KO mice (*Gould et al.,*

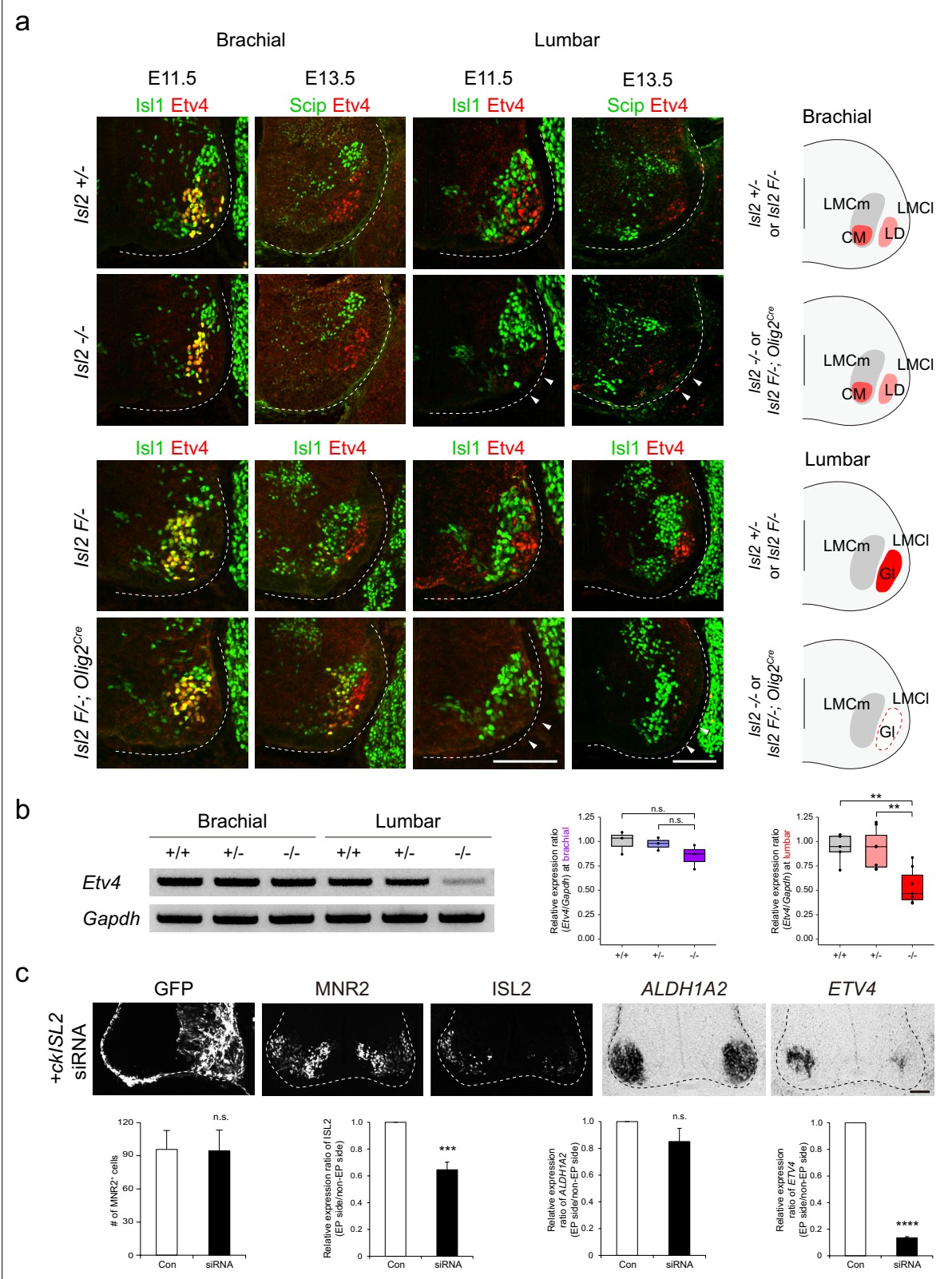

**Figure 5.** Loss of Etv4 expression in *Isl2*-null motor neurons (MNs). (**a**) Expression of Isl1, Scip, and Etv4 in brachial and lumbar spinal cords of *Isl2 +/-*, *Isl2* knockout (KO), *Isl2 F/−*, and *Isl2 F/−*; *Olig2^Cre^* animals at E11.5 and E13.5. Note that Etv4 expression vanishes in lumbar, but not in brachial MNs (arrowheads). Summary diagram depicts the position of lateral motor column (LMC) motor pools innervating the cutaneous maximus (CM), latissimus dorsi (LD), and gluteus (Gl) muscles for each genotype. Scale bars, 100 μm. (**b**) RT-PCR results and quantification demonstrate reduced *Etv4* transcript

*Figure 5 continued on next page*

*Figure 5 continued*

levels in E12.5 *Isl2*-null lumbar spinal cords. Relative expression of *Etv4* at brachial: 1.00±0.06 for *Isl2* +/+, 0.98±0.04 for *Isl2* +/-, and 0.85±0.07 for *Isl2* KO; p=0.2692 for *Isl2* +/+ vs. *Isl2* KO, p=0.3736 for *Isl2* +/- vs. *Isl2* KO; n=3 animals per genotype. Relative expression of *Etv4* at lumbar: 1.00±0.08 for *Isl2* +/+, 0.96±0.08 for *Isl2* +/-, and 0.54±0.07 for *Isl2* KO; p=0.0020 for *Isl2* +/+ vs. *Isl2* KO, p=0.0024 for *Isl2* +/- vs. *Isl2* KO; n=6–8 animals per genotype. Box plots illustrate distribution with median (center line), first and third quartiles (box boundaries), and 10th-90th percentiles (whiskers). One-way ANOVA with Tukey's test; see *Supplementary file 2* and source data for detailed n and statistics. **p<0.01, n.s.=not significant. (**c**) Expression of GFP, MNR2, ISL2, *ALDH1A2*, and *ETV4* in HH stage 25 chick neural tubes, electroporated with siRNA against *ISL2*. Quantification analysis includes the average MNR2-expressing MNs per embryo (96±17 MNs for control, vs. 95±19 for siRNA-electroporated chick, p=0.9622), average fluorescent intensity of ISL2 expression per embryo (1.00 of relative expression ratio for control, vs. 0.65±0.06 for siRNA-electroporated side, p=0.0008), and average relative expression of *ALDH1A2* per embryo (1.00 for control, vs. 0.85±0.10 for siRNA-electroporated side, p=0.3340), and average relative expression of *ETV4* per embryo (1.00 for control, vs. 0.13±0.01 for siRNA-electroporated side, p<0.0001). Note that the *ETV4* transcript was downregulated on the electroporated side (right; n=3–4 animals per group; SEM is shown; unpaired Student's t-test; see *Supplementary file 2* and source data for detailed n and statistics). ****p<0.0001, ***p<0.001, n.s.=not significant. Scale bars, 50 µm.

The online version of this article includes the following source data and figure supplement(s) for figure 5:

**Source data 1.** Quantification of relative expression of *Etv4* in brachial and lumbar spinal cords of wild-type, *Isl2* +/- and *Isl2*-null embryos for *Figure 5b*.

**Source data 2.** Quantification of MNR2[+] cells and relative expression of ISL2, *ETV4*, *ALDH1A2* in chick spinal cords, electroporated with siRNA against *ISL2* for *Figure 5c*.

**Source data 3.** Unedited raw agarose gel pictures and labeled gel pictures of RT-PCR results for *Figure 5b*.

**Figure supplement 1.** Original unprocessed images of RT-PCR results in *Figure 5b*.

**Figure supplement 2.** Assessment of siRNA knockdown (KD) efficiency of siRNA.

**Figure supplement 2—source data 1.** Unprocessed blot and the labeled blot images for reference (*Figure 5—figure supplement 2a–b*).

*2008*). The position and dendritic arborization of Rf motor pools remained unchanged in *Isl2* KO mice (*Figure 7d*). Considering that Etv4 expression has been previously reported to be transient until E13.5 in Rf motor pools, it may not play a significant role in the dendritic arborization of Rf motor pools in adults (*De Marco Garcia and Jessell, 2008*). To investigate the connectivity of Gl motor pools further, we next analyzed the density of sensory synaptic contacts in control and *Isl2* KO mice. Rh-Dex was injected into the Gl muscle at P16, and the number and density of vesicular glutamate transporter 1 (vGluT1), a marker of proprioceptive synapses, were analyzed at P21. We analyzed the synaptic density in two different regions: zone 1, containing normal LMC motor columns, and zone 2, where ectopic MNs reside (*Figure 7e*). In control mice, all Rh-Dex[+] Gl MNs were located in zone 1, while in *Isl2* KO mice, only about 42% were found in zone 1 (*Figure 7e–g*).

Next, we analyzed the synaptic density of vGluT1[+] boutons on motor neuronal somata. Overall, the density of vGluT1 immunoreactivity in zone 1 was higher than in zone 2, which is expected since more MNs reside in zone 1 where more sensorimotor synapses are present. Compared to the control mice, the density of vGluT1 immunoreactivity in *Isl2* KO mice did not change, indicating that overall sensorimotor connection is not affected in them (*Figure 7h*). However, when we analyzed the number of vGluT1[+] boutons in contact with soma of Rh-Dex[+] Gl MNs, we observed a great reduction in the number of boutons in *Isl2* KO mice (*Figure 7i–k*). Furthermore, the number of boutons in zone 2, but not in zone 1, was significantly reduced in *Isl2* KO mice (*Figure 7k*). Thus, ectopically located Gl MNs are more likely to receive fewer proprioceptive sensory inputs compared to the MNs in a normal position in the absence of *Isl2*. Collectively, our findings suggest that *Isl2* KO mice displayed reduced and misoriented dendrites of Gl motor pools, leading to defective sensorimotor connections.

## Aberrant NMJ formation and terminal axon branching in *Isl2* KO mice

To investigate whether the axon projections from mispositioned motor columns successfully reach their target muscles in the limb, we traced the trajectories of LMC axons in *Isl2* KO mice that carried the *Hb9::GFP*. LMC axons reach the base of the limb and diverge to either dorsal or ventral limb trajectories. The major brachial and lumbar plexus appeared mostly normal in *Isl2* KO mice (*Figure 8—figure supplement 1a–c*). These results differ from previous studies involving mice with defects in Gdnf ligands or receptors, where the peroneal nerves were severely affected, resulting in limited or no innervation of the target muscles (*Bonanomi et al., 2012*; *Gould et al., 2008*; *Kramer et al., 2006*). Additionally, in our bulk RNA-seq analysis, we found that the expression levels of Gdnf receptors *Ret* and *Gfra1* was unchanged (–1.13-fold, –1.18-fold each), and their transcripts remained detectable

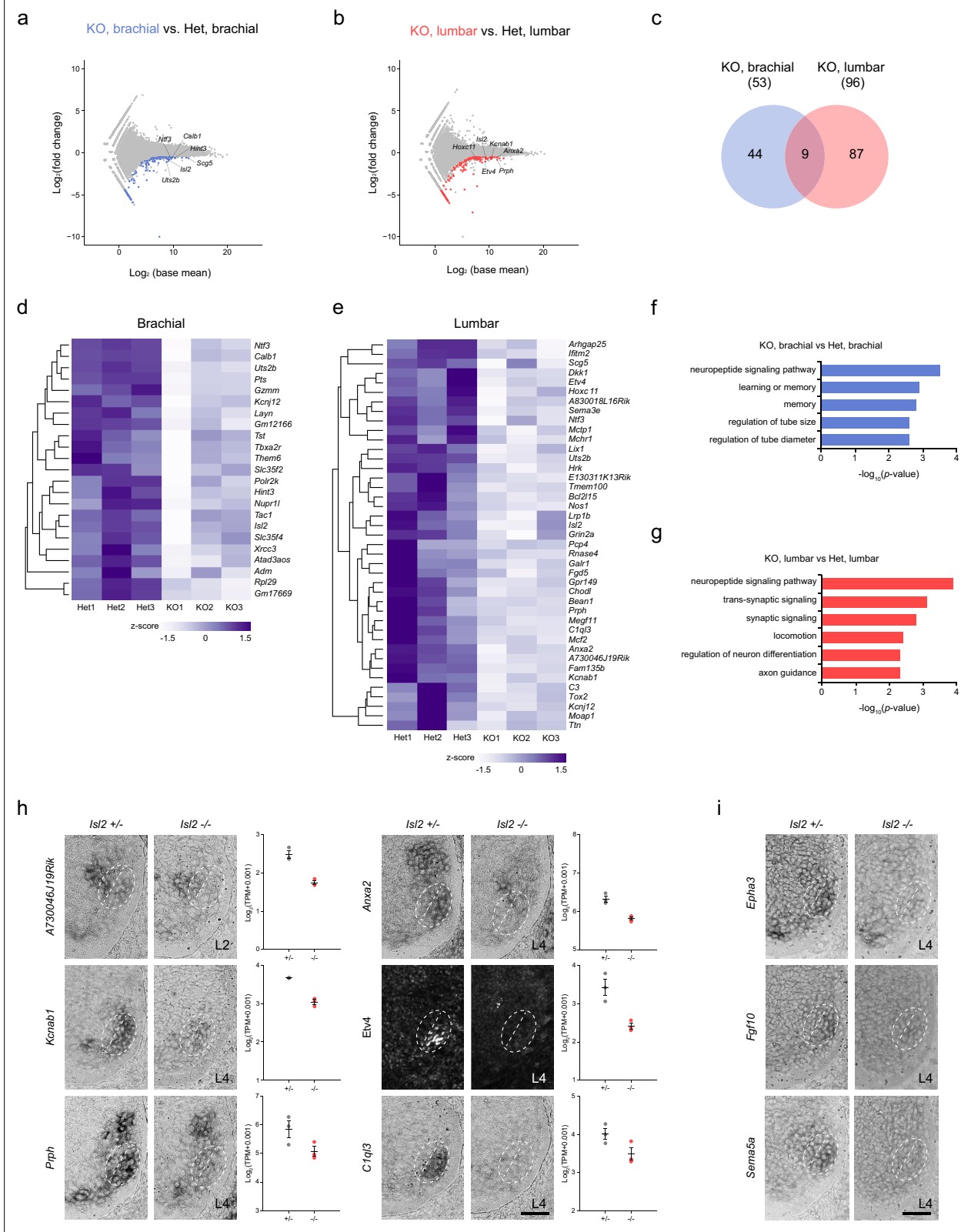

**Figure 6.** Transcriptome analysis of *Isl2*-deficient brachial and lumbar motor neurons (MNs) at E12.5. (**a, b**) MA plots highlighting top differentially expressed genes (DEGs) in brachial and lumbar MNs of E12.5 *Isl2*-null and Isl2 +/- embryos. (**c**) Venn diagram depicting the overlap of downregulated genes in brachial and lumbar MNs in *Isl2*-null embryos. (**d, e**) Heatmaps illustrating selected downregulated genes in the brachial and lumbar spinal cords of *Isl2*-deleted embryos. (**f, g**) Gene ontology (GO) analysis of downregulated DEGs at the brachial and lumbar levels in *Isl2* knockout (KO)

*Figure 6 continued on next page*

*Figure 6 continued*

embryos. (**h**) In situ hybridization and immunohistochemistry of selected downregulated genes in E12.5 *Isl2*-deficient lumbar spinal cords. Dotted lines indicate the position of LMCm and LMCl. Data points of the graphs indicate normalized read counts from individual animals. SEM is shown. (**i**) Expression assessment of *Epha3, Fgf10,* and *Sema5a,* genes enriched in sb.LMCl.2 subcluster from scRNA-seq analysis in *Figure 6—figure supplement 1*. Scale bars, 50 µm.

The online version of this article includes the following figure supplement(s) for figure 6:

**Figure supplement 1.** Identification of lateral motor column (LMC) clusters in which the gene signatures relate to *Isl2*.

in *Isl2*-deficient Gl motor pools when assessed by in situ hybridization analysis (*Figure 8—figure supplement 1d*). Thus, these findings suggest that Gdnf-dependent guidance of LMC projections in the hindlimb may not necessarily depend on Isl2.

We then examined the formation of the NMJ in the hindlimb muscles. Due to the postnatal lethality of *Isl2* cKO, we analyzed the muscles of these mice at E18.5, prior to birth, when NMJ formation is in progress (*Figure 8—figure supplement 2*). GFP-labeled axons arrived in *Isl2*-deficient Gl muscles; however, their terminal arbors were defasciculated and misrouted. Fluorescence intensity of α-bunga-rotoxin (BTX) at the NMJs was greatly reduced in *Isl2* cKO. Furthermore, AChR clustering appeared to be incomplete, showing slender and elongated NMJs, or some NMJs were abnormally concentrated in proximity to the primary axon shaft in *Isl2* cKO (*Figure 8—figure supplement 2b*).

Because NMJs mature during the first postnatal weeks, we next examined the Gl muscles at birth when NMJ formation is ongoing, and at 2 weeks of age when synaptic elimination is almost complete (*Wyatt and Balice-Gordon, 2003*). At P0, Hb9::GFP-labeled main motor axon bundles were compa-rable across *Isl2*-deficient Gl muscles; however, the branches originating from the main bundles and the terminal arbors were drastically diminished. The number of NMJs in Gl was reduced to 84% in the *Isl2* mutant group, and the length and complexity of the secondary branches were signifi-cantly reduced in P0 (*Figure 8a–c*). Higher magnification views revealed that numerous short aberrant sprouts and NMJs had developed near the primary branches in *Isl2*-deficient muscles, which was not observed in the control group (*Figure 8a*). The end plate band area occupied by motor axons was reduced to 55%, and the end plate band area occupied by AChR clusters marked by BTX was reduced to 58% in the Gl muscles of the control group (*Figure 8a, d, and e*). At P14, a drastic loss in AChR clusters and excessive growth of motor axons that bypassed the AChR clusters were observed in *Isl2* KO mice (*Figure 8a, f, and g*). When the few surviving NMJs were visualized during the period of synaptic elimination, a significant portion of nerve terminals were aberrant in *Isl2* KO mice, showing faint AChR clusters, polyinnervation, denervation, and swelling of axons. The area of the NMJ defined by the nerve terminal and the area of AChR clusters were also enlarged in the muscles of *Isl2* KO mice. Fragmented and less compact AChR clusters were more abundant in *Isl2* KO mice than in heterozygote mice (*Figure 8k–r*). In Tfl muscle, abnormal AChRs were formed along axon branches at P0, and a drastic loss in AChR clusters and polyinnervation were observed at P14 and P28 in *Isl2* KO mice (*Figure 8h, i, and j*). However, Rf muscles, whose motor pools transiently express Etv4, showed normal arrangement of NMJs (*Figure 8—figure supplement 3*; *De Marco Garcia and Jessell, 2008*). Other muscles innervated by non-Etv4-expressing motor pools had normal NMJ development (*Figure 8—figure supplement 3*). These findings collectively suggest that the primary role of Isl2 lies in the axon terminal arborization of α-MNs within Etv4-expressing motor pools.

## Impaired hindlimb movement in *Isl2*-deficient mice

To determine whether defective sensorimotor connectivity ultimately caused deficits in hindlimb movements, we examined the behaviors of adult *Isl2* mutant mice. We found that about 10% of *Isl2* KO mice survived after birth, although they were smaller with lower body weight (*Figure 9e*, *Supplementary file 1*). Unexpectedly, however, *Isl2* cKO mice, similar to the *Isl2* KO mice as

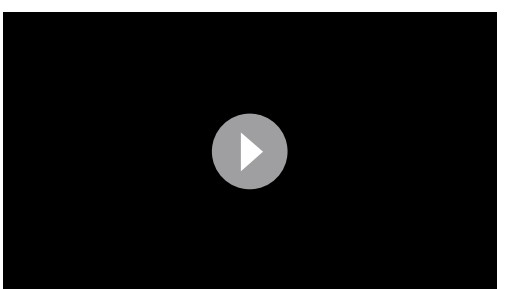

**Video 1.** Example movie of a new born littermate *Isl2*^F/+ and an *Isl2* conditional knockout (cKO) mouse.
https://elifesciences.org/articles/84596/figures#video1

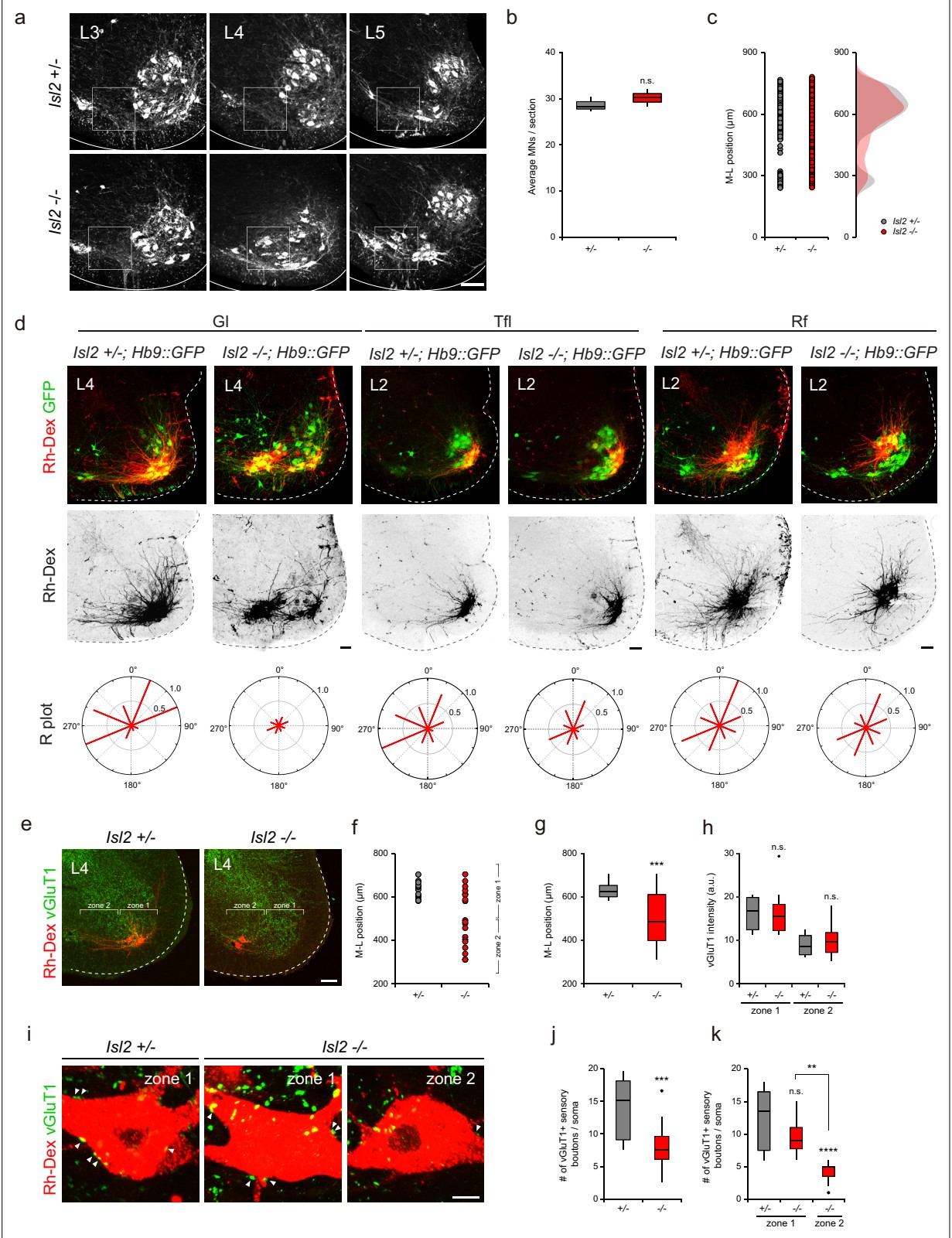

**Figure 7.** Impaired proprioceptive sensory nerve connectivity in the gluteus (Gl) motor pools of *Isl2* mutant mice. (**a–c**) Representative images of adult lumbar spinal cords immunostainined with ChAT antibody and quantification of average number of motor neurons (MNs) (**b**) and their medio-lateral distribution (**c**). (**d**) Representative dendritic arbors of Gl, tensor fasciae latae (Tfl), and rectus femoris (Rf) motor pools retrogradely labeled with rhodamine-dextran (Rh-Dex). Radial (**R**) plots show average dendritic membrane density per octant (red bars) from motor somata (six to eight adjacent

*Figure 7 continued on next page*

*Figure 7 continued*

sections per group, three animals per group). (**e**) Representative images of P21 *Isl2 +/-* and *Isl2*-null lumbar spinal cords with designated zone 1 and zone 2 regions. (**f**) Medio-lateral distribution of Rh-Dex-labeled Gl motor pools in *Isl2 +/-* and *Isl2* KO mice. (**g**) Average medio-lateral position of Gl motor pools in *Isl2 +/-* and *Isl2* knockout mice with box plot depicting the number of dextran-labeled Gl neurons. (**h**) Quantification of vGluT1 intensity in zone 1 and zone 2 in *Isl2 +/-* and *Isl2* KO mice. (**i**) Representative images showing the contact of vGluT1$^+$ Gl sensory boutons with Rh-Dex-labeled Gl MNs (arrowheads). (**j**) The number of sensory boutons per cell body (12.4±1.2 boutons for *Isl2 +/-*, vs. 6.4±0.8 boutons for *Isl2* KO, p=0.0002). (**k**) The number of sensory boutons per cell body in zone 1 and zone 2 (12.4±1.2 boutons for zone 1 in *Isl2 +/-*, 9.5±1.0 boutons for zone 1 in *Isl2* KO, and 4.2±0.5 for zone 2 in *Isl2* KO). p=0.1488 for zone 1 in *Isl2 +/-* vs. zone 1 in *Isl2* KO; p<0.0001 for zone 1 in *Isl2 +/-* vs. zone 2 in *Isl2* KO; p=0.0062 for zone 1 in *Isl2* KO vs. zone 2 in *Isl2* KO. Box plots illustrate distribution with median (center line), first and third quartiles (box boundaries), and 10th-90th percentiles (whiskers).; unpaired Student's t-test for b, g, h, j and one-way ANOVA with Tukey's test for k; see *Supplementary file 2* and source data for detailed n and statistics. ****p<0.0001, ***p<0.001, **p<0.01, n.s.=not significant. Scale bars: (**a**) 100 µm, (**d**) 100 µm, (**e**) 100 µm, (**i**) 10 µm.

The online version of this article includes the following source data for figure 7:

**Source data 1.** Average number of ChAT-expressing motor neurons (MNs) for *Figure 7b*.

**Source data 2.** Quantification of the number of vGluT1$^+$ sensory boutons per cell body and intensity in P21 *Isl2 +/-* and *Isl2* knockout (KO) mice for *Figure 7f–h and j–k*.

previously reported, failed to survive after birth (*Supplementary file 1*, *Video 1*; *Thaler et al., 2004*). Although the discrepancy for the lethality needs to be further investigated, we decided to monitor the postnatal abnormalities using our *Isl2* KO mice. Interestingly, newborn *Isl2* KO mice exhibited rigid hindlimbs that were parted widely, unlike their forelimbs, during free walking (*Video 2*). Adult *Isl2* KO animals also maintained an unnatural extended hindlimb posture when suspended by the tail and during walking (*Figure 9a* and *Video 3*). Footprint analysis showed that *Isl2* KO mice had abnormal gait of the hindlimb and broader width than that of the control group, while the gait of the forelimbs was normal (*Figure 9b and c*). A rigid and broad posture of the hindlimb may be a sign of proximal muscle weakness. Indeed, the muscle mass of the hip muscles, including the Gl and Rf muscles, was significantly reduced in *Isl2* KO animals (*Figure 9d and e*). X-ray imaging and whole skeleton staining with Alizarin red and Alcian blue showed that the overall bone structure and growth were normal in these animals (*Figure 9—figure supplement 1a and b*). We then conducted electromyographic (EMG) analysis to measure hindlimb muscle activity in *Isl2* KO mice. Recordings from the Gl muscles in free-walking *Isl2* KO mice showed lower firing frequency, fewer single-motor-unit potentials, and shorter burst activity duration than the heterozygote animals (*Figure 9f–j*). Taken together, our results show that *Isl2* mutant mice displayed abnormal rigid hindlimb movement, which was accompanied by impaired EMG activity.

## Discussion

The acquisition of specialized motor pools and the specificity of target muscle connections with precision are key steps for effective motor control. We demonstrated that *Isl2* is critical for the clustering of motor pool subsets and the formation of dendritic arborization and NMJs in the hindlimb proximal muscles by demonstrating that *Isl2*-deficient mice display muscle rigidity and abnormal limb coordination. We discuss our findings in the context of genetic programs of Isl2 that direct these connections in a motor pool-specific manner.

### Unraveling motor pool-specific roles of Isl2 and the Gdnf-Etv4 pathway in MN development

Our careful observation of *Isl2* mutant spinal cords revealed that the clustering of MNs was disorganized, with scattered MMC neurons at all segmental levels, and some LMCl motor pools at the hindlimb level. A previous report and also ours showed that development of the PGC population was compromised (*Thaler et al., 2004*). Determining the extent to which altered positions of individual motor neuronal populations contribute to the motor function of *Isl2* mutant mice remains challenging. Nevertheless, our focused investigation on Etv4-expressing motor pools, particularly Gl and Tfl ones, revealed a significant correlation between mislocalized cell bodies, erroneous connectivity, and abnormal hindlimb movements. Previous studies have addressed the importance of motor neuronal cell body position, especially in genetic contexts related to *Hox* pathway, such as Foxp1 and Pbx, which establish neuronal subtype differentiation, organization, and connectivity (*Dasen et al., 2008*;

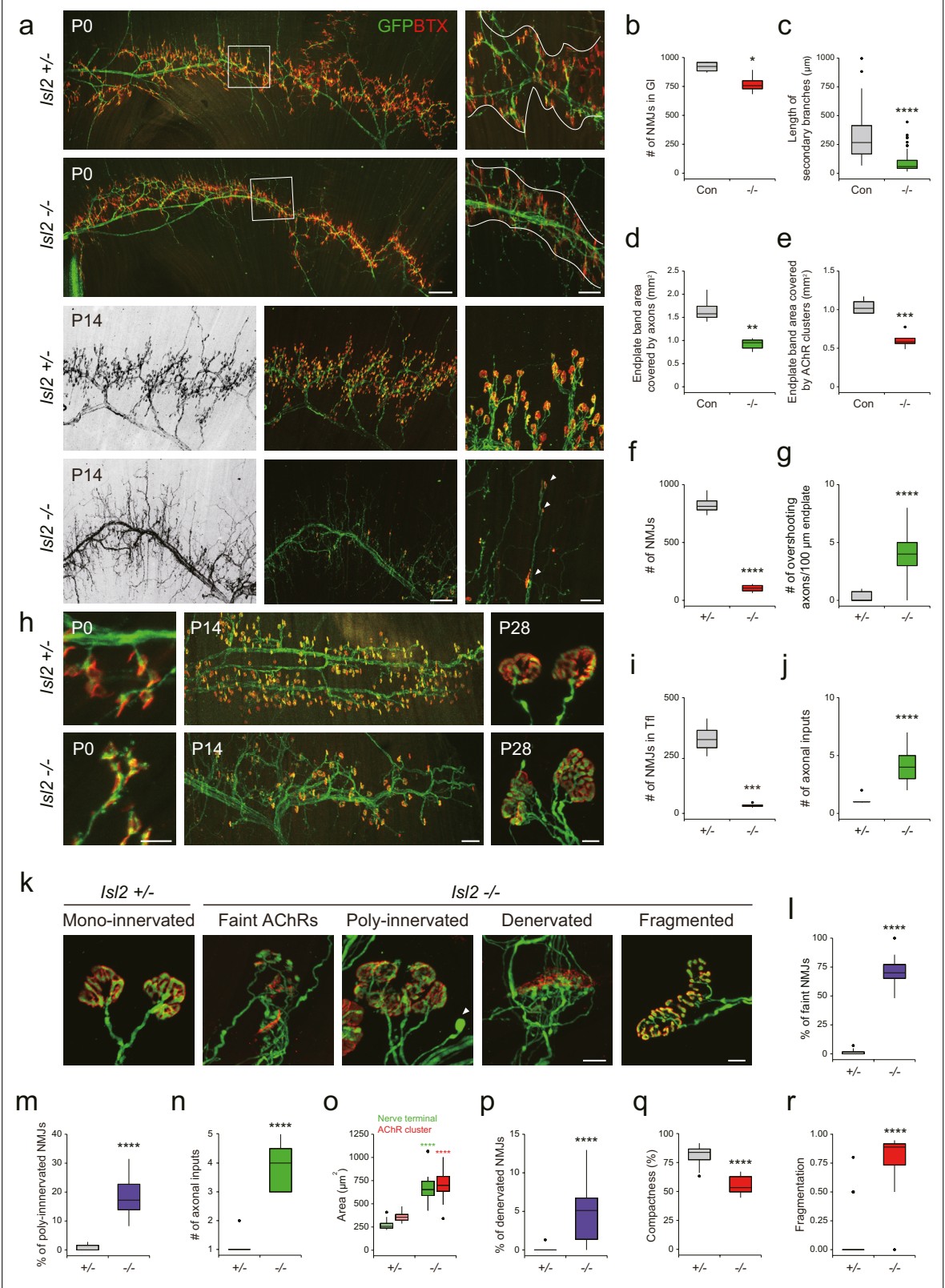

**Figure 8.** Reduced terminal motor axon branching in gluteus (Gl) muscles of *Isl2* mutants. (**a**) Visualization of motor axons and neuromuscular junctions (NMJs) in P0 and P14 Gl muscles with Hb9::GFP (green) and α-bungarotoxin (BTX, red) immunoreactivity. Higher magnification views are provided for the boxed regions. The NMJ region is delineated by white lines in P0. Arrowheads indicate atrophic NMJs on overshooting axons in P14. (**b**) Number of NMJs in P0 Gl muscles (919±23 NMJs for control, vs. 772±44 for *Isl2* knockout [KO], p=0.0257, n=4 muscles per group, three animals). (**c**) Secondary

*Figure 8 continued on next page*

*Figure 8 continued*

branch length (313±17 µm for control, vs. 92±8 µm for *Isl2* KO, p<0.0001, n=90–120 axons per group, three animals) in P0 Gl muscles. (**d**) End plate area covered by motor axons (1.66±0.15 mm$^2$ for control, vs. 0.92±0.06 mm$^2$ for *Isl2* KO, p=0.0014, n=4–5 muscles per group, three animals) in P0 Gl muscles. (**e**) End plate area covered by AChR clusters (1.04±0.05 mm$^2$ for control, vs. 0.61±0.05 mm$^2$ for *Isl2* KO, p=0.0005, n=4–5 muscles per group, three animals) in P0 Gl muscles. (**f**) The number of AChR clusters (828±45 NMJs for *Isl2* +/-, vs. 104±18 for *Isl2* KO, p<0.0001, n=4 muscles per group, three animals) in P14 Gl muscles. (**g**) Axons extending beyond AChR clusters at P14 (0.26±0.07 axons for *Isl2* +/-, vs. 3.92±0.29 for *Isl2* KO, p<0.0001, n=39–46 100 µm intervals of end plate per group, three animals). (**h**) Visualization of motor axons and NMJs in P0, P14, and P28 tensor fasciae latae (Tfl) muscles. (**i**) The number of NMJs at P14 Tfl muscles (324±35 NMJs for *Isl2* +/- vs. 34±5 for *Isl2* KO, p=0.0002, n=4 muscles per group, three animals). (**j**) The number of axonal inputs in P28 Tfl muscles (1.09±0.09 axons for *Isl2* +/-, vs. 4.08±0.40 axons for *Isl2* KO, p<0.0001, n=11–13 NMJs per group, three animals). (**k**) Representative examples of NMJs with morphological defects at higher magnification in Gl muscles at P14. For compactness and fragmentation analysis, P28 NMJs were analyzed. A swelling axon is indicated by an arrowhead. (**l–r**) Percentage of faint NMJs (1.12 ± 0.48% for *Isl2* +/-, vs. 70.47 ± 3.22% for *Isl2* KO, p<0.0001, n=16–17 random fields per group), polyinnervated NMJs (0.81 ± 0.26% for *Isl2* +/-, vs. 18.49 ± 2.75% for *Isl2* KO, p<0.0001, n=8–17 random fields per group), number of axonal inputs (1.05±0.05 axons for *Isl2* +/-, vs. 3.91±0.25 axons for *Isl2* KO, p<0.0001, n=11–22 NMJs per group), area of AChR clusters (357±11 µm$^2$ for *Isl2* +/-, vs. 701±57 µm$^2$ for *Isl2* KO, p<0.0001, n=11–22 NMJs per group) and nerve terminal (270±10 µm$^2$ for *Isl2* +/-, vs. 696±63 µm$^2$ for *Isl2* KO, p<0.0001, n=11–22 NMJs per group), percentage of denervated NMJs (0.08 ± 0.08% for *Isl2* +/-, vs. 4.95 ± 1.04% for *Isl2* KO, p<0.0001, n=16–17 random fields per group), compactness (81.78 ± 1.07% for *Isl2* +/-, vs. 55.52 ± 1.99% for *Isl2* KO, p<0.0001, n=15–39 NMJs per group) and fragmentation (0.06±0.03 for *Isl2* +/-, vs. 0.78±0.06 for *Isl2* KO, p<0.0001, n=15–39 NMJs per group) of AChR clusters were measured. n=3 animals per group; box plots illustrate distribution with median (center line), first and third quartiles (box boundaries), and 10th and 90th percentiles (whisker); unpaired Student's t-test in **b-f**, and **o**; Mann-Whitney test in **g**, **j**, **l-n** and **p-r**; see *Supplementary file 2* and source data for detailed n and statistics. ****p<0.0001, ***p<0.001, **p<0.01, *p<0.05, n.s.=not significant. Scale bars: (**a**) 200 µm (for low-magnification images) and 50 µm (for high-magnification images), (**h**) 20 µm in P0, 100 µm in P14, and 10 µm in P28. (**k**) 10 µm.

The online version of this article includes the following source data and figure supplement(s) for figure 8:

**Source data 1.** Quantification of gluteus (Gl) neuromuscular junctions (NMJs) in *Isl2* +/- and *Isl2* knockout (KO) mice for *Figure 8b–g and l–r*.

**Source data 2.** Quantification of tensor fasciae latae (Tfl) neuromuscular junctions (NMJs) in *Isl2* +/- and *Isl2* knockout (KO) mice for *Figure 8i–j*.

**Figure supplement 1.** Normal motor nerve innervation in *Isl2* mutant hindlimbs.

**Figure supplement 2.** Reduced terminal motor axon branching and immature neuromuscular junctions (NMJs) in gluteus (Gl) muscles of *Isl2* conditional knockout (cKO).

**Figure supplement 3.** Organization of neuromuscular junctions (NMJs) in various hindlimb muscles of *Isl2* mutants.

*Hanley et al., 2016*; *Sürmeli et al., 2011*). In the absence of these genes, MNs intermingle, leading to a variety of developmental defects in sensorimotor circuits and behaviors. Remarkably, expression of representative motor pool markers, including Etv4, was downregulated in *Foxp1*-null mice and was spared in *Pbx*-null mice, implying the involvement of a complex and hierarchical transcriptional network in constructing sensorimotor connectivity in specific motor pools. Herein, we demonstrated that Isl2 plays a crucial role in the correct position and proper function of Etv4-expressing LMC motor pools, particularly at the hindlimb level.

Additional compelling evidence that Isl2 mainly functions in Etv4$^+$ motor pools comes from the striking similarities and differences observed in the mutant phenotypes of Isl2 and the Gdnf-Etv4 pathway, requiring a detailed comparison. Gdnf has been recognized as a critical factor for the survival and innervation of MNs during early MN development (*Henderson et al., 1994*). Deletion of *Gdnf* or its receptor *Ret* or *Gfra1* leads to impaired innervation of the peroneal nerve, resulting in no or disrupted axon innervation in major hindlimb muscles, such as Edl, Atib, psoas major, Gmax, and Tfl, to varying degrees (*Bonanomi et al., 2012*; *Gould et al., 2008*; *Kramer et al., 2006*). In contrast, *Isl2* mutant mice showed normal axon projection of the peroneal nerve and relatively normal axon innervation in most hindlimb muscles, unlike motor axons of mice deficient in *Gdnf* or its receptors. Notably, discrepancies between the roles of Gdnf and Isl2 also emerged in types of motor pools affected. The removal of *Gdnf*, its receptor *Ret* or *Gfra1,* or *Etv4* has been mostly characterized in CM and LD motor pools, the Etv4-expressing motor pools at the forelimb level, but their roles in the lumbar motor pools have not been addressed (*Arber et al., 2000*; *Haase et al., 2002*; *Livet et al., 2002*; *Vrieseling and Arber, 2006*). In our study, we observed a selective loss of *Etv4* expression only in the lumbar spinal cord of *Isl2* KO mice, specifically in Gl and Tfl motor pools. These motor pools displayed a phenotype almost identical to CM and LD motor pools of *Etv4*-null mice, including mispositioned cell bodies, impaired dendrite and axon arborization, defective NMJs, and reduced sensorimotor connectivity. In addition, acute KD of *ISL2* in the chicken neural tube and in MN-specific *Isl2* cKO mice was sufficient to abolish *Etv4* transcripts, suggesting that Isl2 positively regulates *Etv4* expression in a cell-autonomous

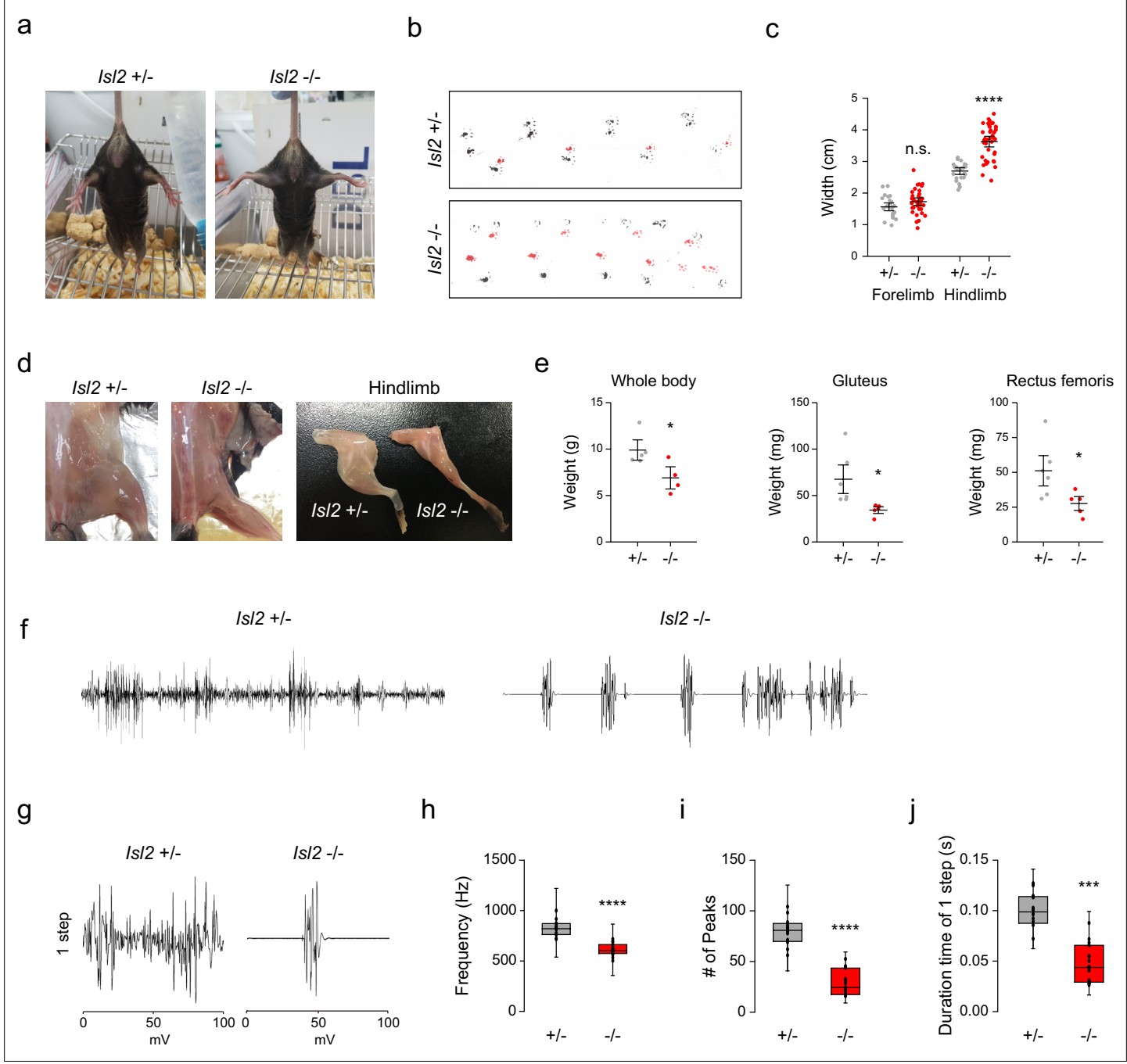

**Figure 9.** Hindlimb gait impairment in *Isl2* mutants. (**a**) Tail suspension test images showing rigid wide-open posture of the hindlimbs in *Isl2* mutant mice. (**b**) Footprint patterns of 3-month-old *Isl2* +/- and *Isl2* knockout (KO) mice. (**c**) Measurement of the stride width of forelimbs (1.56±0.06 cm for *Isl2* +/-, vs. 1.73±0.06 cm for *Isl2* KO, p=0.0782) and hindlimbs (2.70±0.05 cm for *Isl2* +/-, vs. 3.63±0.09 cm for *Isl2* KO, p<0.0001). n=5–8 animals per genotype; SEM is shown; unpaired Student's t-test; see *Supplementary file 2* and source data for detailed n and statistics. ****p<0.0001, n.s.=not significant. (**d**) Gross appearance of lower limbs and dissected hindlimb samples of adult *Isl2* +/- and *Isl2* KO mice. (**e**) Measurement of body (9.9±0.8 g for *Isl2* +/-, vs. 6.9±0.8 g for *Isl2* KO, p=0.0339) and muscle weight for gluteus (Gl) (67.6±12.6 mg for *Isl2* +/-, vs. 34.3±2.7 mg for *Isl2* KO, p=0.0316, n=5–6 muscles per group) and rectus femoris muscles (51.1±8.2 mg for *Isl2* +/-, vs. 27.6±3.8 mg for *Isl2* KO, p=0.0384, n=5–6 muscles per group). n=5 animals for heterozygote, n=4 animals for KO; SEM is shown; unpaired Student's t-test; see *Supplementary file 2* and source data for detailed n and statistics. *p<0.05. (**f**) Representative electromyographic (EMG) recordings of Gl muscles during free-walking. (**g**) Signature of individual footsteps. (**h**–**j**) Quantification of average frequency (861±18 Hz for *Isl2* +/-, vs. 619±16 Hz for *Isl2* KO, p<0.0001), number of single-motor-unit potentials (80±4 peaks for *Isl2* +/-, vs. 42±3 peaks for *Isl2* KO, p<0.0001), and burst activity duration (0.096±0.005 s for *Isl2* +/-, vs. 0.068±0.005 s for *Isl2* KO, p=0.0001) in EMG recordings of Gl muscles. Three animals per group; box plots illustrate distribution with median (center line), first and third quartiles (box boundaries),

*Figure 9 continued on next page*

*Figure 9 continued*

and 10th-90th percentiles (whiskers); unpaired Student's t-test; see ***Supplementary file 2*** and source data for detailed n and statistics. ****p<0.0001, ***p<0.001.

The online version of this article includes the following source data and figure supplement(s) for figure 9:

**Source data 1.** Measurement of stride width of forelimbs and hindlimbs in footprint analysis of 3-month-old *Isl2* +/- and *Isl2* knockout (KO) mice for ***Figure 9c***.

**Source data 2.** Quantification of the average frequency, number of single-motor-unit potentials, and burst activity duration in electromyographic (EMG) recordings of the gluteus (GI) muscles for ***Figure 9h–j***.

**Figure supplement 1.** Skeletal analysis in *Isl2* mutant mice.

**Figure supplement 1—source data 1.** Quantification of tibia length in X-ray analysis for ***Figure 9—figure supplement 1b***.

---

manner. Thus, Gdnf likely plays broader roles in guiding most LMC projections and also promoting differentiation of Etv4-expressing motor pools at both limb levels. In contrast, Isl2 plays a precise role in fine-tuning the position of MNs, including MMC and LMCl, and also functions in selected lumbar motor pools at the hindlimb, particularly in establishing sensorimotor connectivity during the late period of development.

Nevertheless, Etv4 expression in the CM motor pool at the brachial level was intact in both *Isl2* KO mice and cKO mice, implying that another complex genetic program works in conjunction with Isl2 to contribute to segment-specific regulation. For instance, multiple *Hox* genes control Gdnf signaling, with *Hoxc6* activating *Ret* and *Gfra1* expression in rostral brachial spinal cords, regulating terminal arborization of CM and LD motor pools, while *Hoxc8* activates *Ret* and *Gfra3* expression in caudal brachial spinal cords, controlling limb innervation of distal nerves (*Catela et al., 2022*). Similar regulatory pathways involving Hox/Meis in *Ret* expression have been suggested at lumbar levels, with misexpression of *Hoxc10* or *Hoxd10* in chicken embryos resulting in *Ret* expression arising in the neural tube (*Catela et al., 2022*). Our bulk RNA-seq analysis identified *Hoxc11* as one of the DEGs, indicating a potential compromise of proper lumbar identity in the absence of Isl2. However, the expression of *Ret* and *Gfra1* was not changed in *Isl2* KO mice when assessed by bulk RNA-seq analysis and in situ hybridization, and overall hindlimb motor projections remained intact. Although further investigation is necessary to fully elucidate the hierarchy among Hox, Gdnf, and Isl2, it is evident that each factor plays a crucial role, to varying degrees, in the proper expression of *Etv4* within proximal hindlimb motor pools.

Thaler et al. initially reported that *Isl2* KO mice died within a day after birth due to respiratory failure as well as MN developmental defects (*Thaler et al., 2004*). In contrast, we found that the identical mouse line exhibited a greater survival rate, about 10%. On the other hand, MN-specific *Isl2* cKO mice showed similar lethality, with all pups dying on the first day after birth. There are several possibilities to explain the variable mortality among different mouse lines lacking *Isl2*. First, acute elimination of *Isl2* gene in developing MNs could be more detrimental, minimizing potential developmental compensation through altered expression of other genes reconciling the impact of the target KO. Second, the genetic background of the mice may play a role, as it has been reported that different genetic backgrounds can cause varying phenotypes and lethality in KO mouse models (*El-Brolosy and Stainier, 2017*). For instance, the

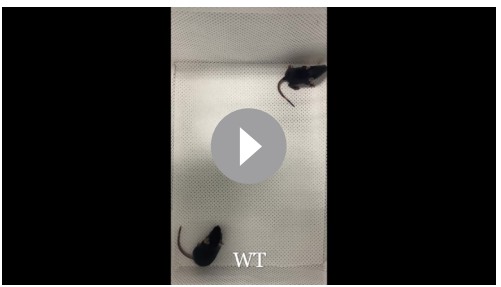

**Video 2.** Example movie of a P14 littermate control *Isl2* +/- mouse and an *Isl2* knockout (KO) mouse during walking.

https://elifesciences.org/articles/84596/figures#video2

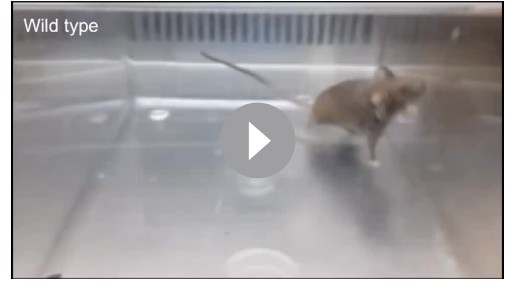

**Video 3.** Example movie of an adult littermate *Isl2* +/- and an *Isl2* knockout (KO) mouse during walking.

https://elifesciences.org/articles/84596/figures#video3

lethality of Lhx4 embryos was found to be different in C57BL/6J relative to the mixed background (*Gergics et al., 2015*). Similarly, the genetic background of original *Isl2* global null mice could be different from ours, with these mice being occasionally backcrossed onto the C57BL/6 genetic background. Despite the variable survival rates, the defects found in *Isl2*-null and conditional KO embryos were essentially identical, including scattered lumbar motor pools, absence of Etv4 expression, and defective NMJs, all of which are closely related to the adult behavioral phenotypes found in global *Isl2* KO mice. These findings further highlight the critical role of Isl2 in MN development and locomotion. Overall, our study provides a comprehensive analysis of the broad spectrum of phenotypes resulting from removal of *Isl2* expression.

## Transcriptomic analysis to uncover Isl2-dependent genes for motor pool formation

Recent genome-wide transcriptomic studies have shed light on dynamic transcriptional changes occurring during the development of central nervous system, including spinal MNs (*Amin et al., 2021*; *Delile et al., 2019*; *Liau et al., 2021*; *Wang et al., 2022*). However, the molecular signature responsible for the organization of multiple individual motor pools has remained elusive. In this study, we investigated the detailed spatiotemporal expression of Isl2 among multiple lineages of MNs using a published scRNA-seq dataset. We successfully segregated the clades of LMC motor pools into six subclusters that encompassed most of the major motor pools at the rostral lumbar segment. Our investigation revealed that sb.LMCl.2 exhibits a molecular signature corresponding to Etv4-expressing Gl, Tfl, and Rf motor pools, with Isl2 expression relatively enriched in this subcluster. Similar LMC subclusters expressing *Etv4* was also identified by other groups sharing similar molecular codes (*Liau et al., 2023*; *Wang et al., 2022*).

In the current study, we aimed to integrate bulk and scRNAseq analyses datasets to narrow down the motor pools whose terminal differentiation is primarily dependent on the presence of Isl2, as well as to identify potential target genes under the control of Isl2. From the *Isl2*-deficient lumbar spinal cords, we identified 96 DEGs downregulated, and 87 of them were lumbar-specific. In particular, seven genes (*C1ql3*, *Etv4*, *A830018L16Rik*, *Barx2*, *Kcnab1*, *Hrk,* and *Prph*) were enriched genes in subcluster sb.LMCl.2, and five of them (*C1ql3*, *Etv4*, *A830018L16Rik*, *Barx2*, and *Kcnab1*) belonged to the top 30 of them (see *Figure 6—figure supplement 1i*, *Supplementary file 3b-c*). Upon comparing our DEGs with another scRNA-seq dataset analyzed by *Liau et al., 2023* we found similar results. Ten genes (*Anxa2*, *A730046J19Rik*, *Chodl*, *C1ql3*, *Etv4*, *Kcnab1*, *Lix1*, *Megf11*, *Prph*, and *Uts2b*) were also found to be part of the molecular features characterizing the LMCl subcluster called c/rl4 (*Lhx1*[+] *Etv4*[+]), confirming the validity of our analyses (*Liau et al., 2023*). We next aimed to determine whether our approach allowed us to discover new marker genes for designated motor pools. Among the list of enriched genes in sb.LMCl.2 retrieved from scRNA-seq analysis, we demonstrated that three genes *Sema5a*, *Epha3,* and *Fgf10,* exhibited selective expression in Gl motor pools and reduced expression in *Isl2*null mice according to our in situ hybridization results. Overall, our comprehensive approach, combining bulk and scRNA-seq analyses datasets, has proved highly valuable in identifying new potential motor pool markers and unraveling the genetic program underlying their development.

Our transcriptomic analysis and in vivo validation of candidate targets revealed that numerous genes involved in motor neuronal differentiation and diseases were indeed under the control of Isl2. Among the 96 DEGs downregulated in *Isl2* mutant spinal cords, 32 genes (i.e., *Adra2c*, *Arhgap25*, *C1ql3*, *C3*, *Cck*, *Chrm5*, *Dkk1*, *Etv4*, *Fstl3*, *Gabra5*, *Gpr149*, *Grin2a*, *Hrk*, *Htr4*, *Kcnab1*, *Kcnj12*, *Mcf2*, *Mchr1*, *Mctp1*, *Nos1*, *Ntf3*, *Oprd1*, *P2rx4*, *Pcp4*, *Pnmt*, *Prph*, *Pvalb*, *Scnn1a*, *Sema3e*, *Tmem100*, *Tmsb15b2*, and *Tnc*) were associated with axon- or synapse-related GO terms, or their protein expression was found in axons or synapses (*Dhaese et al., 2009*; *Fink et al., 2017*; *Martinelli et al., 2016*; *Molinard-Chenu et al., 2020*; *Wu et al., 2019*). In addition, 14 genes (i.e., *Anxa2*, *Bcl2l15*, *C3*, *Chodl*, *Fam135b*, *Hrk*, *Lix1*, *Lrp1b*, *Moap1*, *Pcp4*, *Prph*, *Rnase4*, *Scg5*, and *Scnn1a*) were reported as ALS- or MN disease-associated genes in the literature (*Benoit et al., 2013*; *Ghavami et al., 2014*; *Gros-Louis et al., 2004*; *Lederer et al., 2007*; *Li et al., 2013*; *Orr et al., 2020*; *Sasongko et al., 2010*; *Sheila et al., 2019*; *Shu et al., 2022*; *Sleigh et al., 2014*; *Su et al., 2022*; *Wei et al., 2011*; *Wu et al., 2019*). Therefore, approximately 40% of *Isl2*-dependent DEGs were involved in the maturation and pathological processes of MNs. *Prph*, a neuronal intermediate filament protein implicated in ALS, showed the largest fold changes (<−0.5 log$_2$-fold change) in our transcriptomic analysis and was selectively

downregulated in specific motor pools in *Isl2* mutant mice. In addition, earlier extinction of *Prph* in developing MNs correlated with progressive degeneration of spinal MNs and NMJs, which indicated that the defects observed in *Isl2* mutant mice are relevant to early-onset progressive MN disorders such as SMA and ALS (*Amin et al., 2015*; *Amin et al., 2021*; *Beaulieu and Julien, 2003*; *Chen et al., 2013*; *Gros-Louis et al., 2004*; *Sabbatini et al., 2021*). C1ql3 has been shown to promote synapse formation and maintenance in certain excitatory neurons, and its absence results in fewer excitatory synapses with diverse behavioral defects (*Martinelli et al., 2016*). Given that these MN disease-related genes were all selectively downregulated in specific motor pools that showed deteriorating features in the absence of *Isl2*, it is likely that proper regulation of these genes by Isl2 ensures proper MN development, whereas dysregulation leads to MN degeneration.

In summary, our findings suggest that pool-specific Isl2 activity is important for the fidelity of motor pool organization and motor circuit connectivity for hindlimb locomotion. Furthermore, Isl2 is responsible for the transcriptional control of a variety of genes involved in MN differentiation, axon development, and synaptic organization, and its absence may give rise to a pathological condition of MNs.

# Materials and methods

## Key resources table

| Reagent type (species) or resource | Designation | Source or reference | Identifiers | Additional information |
|---|---|---|---|---|
| Strain, strain background (Mouse, female) | C57BL/6J | Damul Science Co. | | |
| Genetic reagent (*Mus musculus*) | *Isl2*$^{-/-}$ | PMID:14766174 | MGI:3046260 | |
| Genetic reagent (*M. musculus*) | *Isl2*$^{flox/flox}$ | This paper | MGI:109156 | See Materials and methods, and *Figure 3—figure supplement 1a* |
| Genetic reagent (*M. musculus*) | Tg(Mnx1-GFP)1Slp | PMID:15201216 | MGI:3767834 | |
| Genetic reagent (*M. musculus*) | B6.Cg-Tg(Thy1-YFP)16Jrs/J | Jackson Laboratory PMID:11086982 | RRID:IMSR_JAX:003709 | |
| Genetic reagent (*M. musculus*) | Olig2$^{tm1(cre)Tmj}$ | PMID:18046410 | MGI:3774124 | |
| Cell line (human) | 293[HEK-293] | Korean Cell Line Bank | KCLB# 21573 | Verified using STR profiling and confirmed to be mycoplasma-free |
| Antibody | Anti-Olig2 (guinea pig polyclonal) | Jessell lab | | (1:8000) |
| Antibody | Anti-Hb9 (rabbit polyclonal) | PMID:10471502 | | (1:8000) |
| Antibody | Anti-Hb9 (guinea pig polyclonal) | PMID:10482235 | | (1:2000) |
| Antibody | Anti-GFP (mouse monoclonal) | Sigma | Cat# G6539 | (1:2000) |
| Antibody | Anti-Isl1/2 (rabbit polyclonal) | PMID:7528105 PMID:8565076 | | (1:5000) |
| Antibody | Anti-Isl2 (mouse monoclonal) | Santa Cruz | Cat# sc-390746 | (1:500) |
| Antibody | Anti-Lhx3 (guinea pig polyclonal) | Pfaff lab | | (1:4000) |
| Antibody | Anti-ChAT (goat polyclonal) | Chemicon | Cat# AB144P | (1:100) |
| Antibody | Anti-Foxp1 (rabbit polyclonal) | Abcam | Cat# ab16645 | (1:5000) |
| Antibody | Anti-Nkx6.1 (goat polyclonal) | R&D Systems | Cat# AF5857 | (1:1000) |
| Antibody | Anti-Isl1 (goat polyclonal) | R&D Systems | Cat# AF1837 | (1:1000) |
| Antibody | Anti-Isl2 (guinea pig polyclonal) | PMID:14766174 | Cat# AF1837 | (1:8000) |
| Antibody | Anti-Lhx1 (rabbit polyclonal) | Abcam | Cat# ab14554 | (1:500) |
| Antibody | Anti-Scip (guinea pig polyclonal) | Dasen lab | | (1:8000) |

*Continued on next page*

*Continued*

| Reagent type (species) or resource | Designation | Source or reference | Identifiers | Additional information |
|---|---|---|---|---|
| Antibody | Anti-Etv4 (rabbit polyclonal) | Dasen lab | | (1:16,000) |
| Antibody | Anti-Tetramethylrhodamine (rabbit polyclonal) | Invitrogen | Cat# A-6397 | (1:1000) |
| Antibody | Anti-vGluT1 (guinea pig polyclonal) | Sigma | Cat# AB5905 | (1:32,000) |
| Antibody | Anti-nNOS (rabbit polyclonal) | DiaSorin | | (1:16,000) |
| Antibody | Anti-pSMAD (rabbit monoclonal) | Cell Signaling Technology | Cat# 9516S | (1:500) |
| Antibody | Anti-GFP (rabbit polyclonal) | Abcam | Cat# ab290 | (1:1000) |
| Antibody | Anti-Tubulin alpha (rat monoclonal) | AbD Serotec | Cat# MCA77G | (1:10,000) |
| Antibody | Anti-HA (mouse monoclonal) | Covance | Cat# MMS-101R | (1:5000) |
| Antibody | Anti-Isl2 (mouse monoclonal) | DSHB | Cat# 51.4H9 | (1:200) |
| Antibody | Anti-MNR2 (mouse monoclonal) | DSHB | Cat# 81.5C10 | (1:500) |
| Antibody | Anti-Digoxigenin-AP, Fab fragments (sheep polyclonal) | Roche | Cat# 11093274910 | (1:5000) |
| Sequence-based reagent | *ckISL2* siRNA-461 | This paper | | See Materials and methods, and *Figure 5c*, *Figure 5—figure supplement 2a–b* |
| Sequence-based reagent | *ckISL2* siRNA-605 | This paper | | See Materials and methods, and *Figure 5c*, *Figure 5—figure supplement 2a–b* |
| Peptide, recombinant protein | SP6 RNA polymerase | Roche | Cat# 10810274001 | |
| Peptide, recombinant protein | T7 RNA polymerase | Roche | Cat# 10881775001 | |
| Peptide, recombinant protein | Protector RNase Inhibitor | Roche | Cat# 3335399001 | |
| Peptide, recombinant protein | RQ1 RNase-Free DNase | Promega | Cat# M6101 | |
| Peptide, recombinant protein | Proteinase K | VWR | Cat# E195-5ML | |
| Commercial assay or kit | NucleoSpin RNA XS | MACHEREY-NAGEL | Cat# 740902.50 | |
| Commercial assay or kit | pGEM-T Easy Vector Systems | Promega | Cat# A1360 | |
| Commercial assay or kit | DiaStar OneStep RT-PCR kit | Solgent | Cat# DR61-K050 | |
| Commercial assay or kit | DIG RNA Labeling Mix | Roche | Cat# 11277073910 | |
| Chemical compound, drug | Dextran, Tetramethylrhodamine, 3000 MW | Invitrogen | Cat# D3308 | |
| Chemical compound, drug | Alpha-Bungarotoxin Conjugates | Invitrogen | Cat# B35451 | |
| Chemical compound, drug | NBT/BCIP stock solution | Roche | Cat# 11681451001 | |
| Software, algorithm | Zen | ZEISS | Zen Black 2.3 SP1 | https://www.zeiss.com/microscopy/us/products/microscope-software/zen.html |
| Software, algorithm | FV31S-SW | Olympus | 2.5.1.228 | https://www.olympus-lifescience.com/ |

*Continued on next page*

*Continued*

| Reagent type (species) or resource | Designation | Source or reference | Identifiers | Additional information |
| --- | --- | --- | --- | --- |
| Software, algorithm | GraphPad Prism | GraphPad software | Version 9 | https://www.graphpad.com/ |
| Software, algorithm | ImageJ | National Institutes of Health | 1.53k | |
| Software, algorithm | R v3.6.0 | R-project | | https://www.r-project.org/ |
| Software, algorithm | R v4.0.2 | R-project | | https://www.r-project.org/ |
| Software, algorithm | Seurat v3.2.3 | R-project | PMID:29608179 | https://satijalab.org/seurat/get_started.html |
| Software, algorithm | Metascape | PMID:30944313 | | https://metascape.org/ |
| Software, algorithm | Flaski | | | https://flaski.age.mpg.de. |

## Mice

The *Isl2*-null, *Hb9::GFP* and *Thy1::YFP* mice used in this study have been previously described (Jackson laboratory) (*Feng et al., 2000*; *Thaler et al., 2004*). Wild-type C56BL/6 mice (6–8 weeks of age) were purchased from Damul Science Co. (Daejon, Korea). To selectively delete *Isl2* from MNs, we generated a novel *Isl2* flox mouse (Cyagen, China). The flox strategy was designed by Cyagen using the following procedure. The *Isl2* gene (NCBI reference sequence: NM_027397; Ensemble: ENSMUSG00000032318) was located on chromosome 9 in mice. Exons 2–4 were selected as cKO regions. In the targeting vector, the Neo cassette was flanked by SDA sites, and DTA was used for negative selection. C57BL/6N ES cells were used for gene targeting. The KO allele was obtained after specific Cre-mediated recombination. Genotyping assays were designed by Cyagen. First-generation mice were heterozygous for *Isl2* flox expression. Heterozygous mice were bred to generate mice with homozygous *Isl2* flox expression. The *Isl2* flox mouse line was maintained as a homozygous line for breeding with Cre strains. For the Cre line, an MN-specific *Olig2$^{Cre}$* line was used (*Dessaud et al., 2007*; *Kong et al., 2015*). To obtain *Isl2$^{F/KO}$*; *Olig2$^{Cre}$* mice, male *Isl2 +/-*; *Olig2$^{Cre}$* mice were crossed with female *Isl2$^{F/F}$* mice. The *Isl2$^{F/KO}$* or *Isl2$^{F/+}$*; *Olig2$^{Cre}$* mice from the same litters were used as normal controls. For genotyping PCR, genomic DNA was isolated from mouse ear punches digested in DirectPCR (tail) (VIAGEN) and Proteinase K (New England Biolabs). The following primers were used for genotyping PCR: 5'-TGG GAC TAC GGG GTT GTA CTT-3' and 5'-GTT CTG GAG AGC AAG TTG GGA AT-3' to detect a wild-type allele (274 bp) and a flox allele (410 bp). All experiments used protocols approved by the Animal Care and Ethics Committees of the Gwangju Institute of Science and Technology (GIST). The day when a vaginal plug was detected was designated as embryonic day 0.5 (E0.5).

## Immunohistochemistry and in situ hybridization

Immunohistochemistry and in situ hybridization were performed as described previously (*Song et al., 2009*). The following antibodies were used: guinea pig anti-Olig2 (*Wichterle et al., 2002*), rabbit and guinea pig anti-HB9 (*Harrison et al., 1999*), rabbit anti-GFP (Invitrogen), mouse anti-GFP (Sigma), rabbit anti-Isl1/2 (*Pfaff et al., 1996*), mouse anti-Isl2 (Santa Cruz), guinea pig anti-Lhx3 (*Thaler et al., 2004*), goat anti-ChAT (Chemicon), rabbit anti-Foxp1 (Abcam), goat anti-Nkx6.1 (R&D Systems), goat anti-Isl1 (R&D Systems), guinea pig anti-Isl2 (*Thaler et al., 2004*), rabbit anti-Lhx1 (Abcam), guinea pig anti-Scip (*Dasen et al., 2005*), rabbit anti-Etv4 (*Dasen et al., 2005*), rabbit anti-Tetramethylrhodamine (Invitrogen), guinea pig anti-vGluT1 (Millipore), rabbit anti-nNOS (Diasorin), and rabbit anti-pSMAD (Cell Signaling Technology). For NMJ staining, muscles were harvested and were prepared for immunostaining as whole-mount samples. Samples were immunostained with rabbit anti-GFP (Abcam) and Alexa Fluor 555 α-BTX (Invitrogen). For in situ hybridization, embryonic mouse cDNA at E12.5 was used to generate riboprobes for *Isl2*, *Ret*, *Gfra1*, *A730046J19Rik*, *Anxa2*, *Kcnab1*, *Prph*, *C1ql3*, *Epha3*, *Fgf10*, and *Sema5a*. HH stage 25 chicken embryonic cDNA was used to generate riboprobes for chicken *ALDH1A2* and *ETV4* using a one-step RT-PCR kit (Solgent).

## Western blot analysis

293[HEK-293] cells (KCLB No. 21573) were purchased from Korean Cell Line Bank. The cell lines were verified using STR profiling and confirmed to be mycoplasma-free. 293[HEK-293] cells were cultured and transfected with an expression plasmid containing N-terminal truncated chick *ISL2* (aa 38–356) with an HA tag and *ISL2* siRNA using Lipofectamine (Invitrogen). After 36 hr, the cells were dissociated and lysed in lysis buffer for western blotting. The western blotting procedure followed previous protocols (*Song et al., 2009*). Rat anti-α-tubulin (AbD Serotec) and mouse anti-HA (Covance) antibodies were used.

## Chick in ovo electroporation

In ovo electroporation was performed as described previously (*Kim et al., 2016*). A DNA solution was injected into the lumen of the spinal cord of 10–12 chicken embryos at the HH stage, and the embryos were harvested at HH stage 25. To knock down *ISL2*, *ISL2* siRNAs (siRNA-461 sense 5′-GGA CGG UGC UGA ACG AGA A-3′, siRNA-461 antisense 5′-UUC UCG UUC AGC ACC GUC C-3′; siRNA-605 sense 5′-GCU GCA AGG ACA AGA A-3′, siRNA-605 antisense 5′-UUC UUC UUG UCC UUG CAG C-3′) were electroporated. Harvested chicken embryos were processed for immunohistochemistry and in situ hybridization. Rabbit anti-GFP (Invitrogen), mouse anti-Isl2 (DSHB), and mouse anti-MNR2 (DSHB) antibodies were used. Riboprobes for chicken *ALDH1A2* and *ETV4* were amplified from total RNA extracted from HH 25 chicken embryos using a one-step RT-PCR kit (Solgent). For image analysis, consecutive sections at the lumbar level were collected from each embryo. In each ventral quadrant, the number of MNs was counted, and the signal intensity was measured using the ImageJ software (NIH). The intensity values of the electroporated side were normalized to the value of the nonelectroporated side.

## Intramuscular injections of tracers

To label MN dendrites, the target muscles were injected with Rh-Dex (3000 MW, Invitrogen). In brief, P4 mice were anesthetized on ice, and the muscles were exposed by making a small incision in the skin. Approximately 2 μl of 10% Rh-Dex was injected at a single site, and the skin was sutured. After 3 days, the spinal cords were analyzed for dextran labeling, and the muscles were isolated to check the injection site. Analyses of dendritic arborization patterns of the Gl motor pools primarily focused on the L3 and L4 spinal segmental levels. Free-floating 80-μm-thick transverse sections were immunostained for further analysis.

For sensory bouton analysis, P16 mice were anesthetized, and the skin was incised to expose the muscle of interest. Muscles were injected at multiple sites with 10% tetramethylrhodamine dextran (3000 MW, Invitrogen) using a pulled glass microelectrode. At P21, the spinal cords were harvested and processed for immunostaining.

## EMG analysis

For EMG analysis, we used multistranded, Teflon-coated, annealed stainless steel wire (A-M systems, Cat. No. 793200) electrodes that were soldered to an IC socket. After mice were anesthetized with avertin, we implanted the electrodes subcutaneously into the middle of the left and right Gl muscles and secured the connector to the head. The mice were allowed to recover in their cages for at least 2 days before recording began. After the recovery period, we placed the animals in an open field where they could move freely. During locomotor activity, EMG signals were digitized and stored using Clampex 9.2 data acquisition software, while video recordings were made to monitor walking movement for 30 min. EMG and video data were recorded simultaneously. The EMG data was bandpass filtered at 200 Hz to 1 kHz using Clampfit 10.3. We measured and analyzed the number, frequency, and duration of EMG peaks over 1 s.

## Tail suspension test and gait analysis

For the tail suspension test, we suspended each mouse above the cage by its tail and captured its posture on a camera. In the gait analysis, we used 3-month-old *Isl2* KO and littermate female controls. We applied nontoxic water-based paints (red on forepaws and black on hindpaws) to the mouse paws and allowed the mice to walk on white paper. We performed three consecutive trials and measured the stride widths of the forelimbs and hindlimbs.

## RT-PCR

E12.5 embryonic spinal cords were harvested in ice-cold PBS and total RNA was extracted using NucleoSpin RNA XS (MACHEREY-NAGEL). We used 10 ng of total RNA per 15 µl reaction for the RT-PCR with the following primers: (*Gapdh*, 5'-GGA GAA ACC TGC CAA GTA TGA-3', 5'-CCT GTT GCT GTA GCC GTA TT-3', *Etv4* 5'-GGT GAT GGA GTG ATG GGT TAT G-3', 5'GCC TGT CCA AGC AAT GAA ATG-3'). The PCR products were amplified using a one-step RT-PCR kit (Solgent). Band densitometry was quantified for *Etv4* and *Gapdh* using the ImageJ software.

## Image acquisition

For data analysis, images were captured using epifluorescent LSM780 (Zeiss) and FV3000RS (Olympus) confocal microscopes using the Axiovision, ZEN software (Zeiss) and the FV31S software (Olympus).

## Radial plot

To analyze dendritic radiality, motor pools were subdivided into eight divisions, with the center aligned with the center of the motor pool. For each octant, the mean pixel intensity of dendrites, excluding the cell bodies, was quantitated using the ImageJ software and normalized to the octant with the highest value.

## Contour and density plots

MN soma coordinates were acquired with respect to the midline using ImageJ software. To assign x and y coordinates, we normalized experimental variations from different spinal cord sizes, shapes, and sections. Contour, spatial, and density plots were created using the 'ggplot2' package in R-4.0.5. Contour plots were calculated using 'MASS::kde2d()' by performing a 2D kernel density estimation and displaying the results with contours for the distribution of neuron positions.

## Quantification of sensory boutons

For quantitative analysis of vGluT1$^+$ sensory boutons in Rh-Dex$^+$ Gl motor pools, spinal cords were sectioned at a thickness of 40 µm. A z-series of 0.74 µm optically scanned confocal images using a 30× objective were acquired for quantitative analysis. The number of vGluT1$^+$ puncta within each somatic compartment was quantified across the Rh-Dex$^+$ MNs. vGluT1 intensity was determined using ImageJ software, and vGluT1$^+$ synapses were counted manually (*Wang et al., 2019*).

## Quantification of NMJ morphology and imaging

NMJ images were acquired on LSM780 (Zeiss) and FV3000RS (Olympus) confocal microscopes. To analyze NMJs, maximum intensity projections were created using ZEN (Zeiss) and FV31S (Olympus) software. Low-magnification images were acquired using 10× and 20× objectives. For NMJ morphology assessment, confocal settings were optimized: 640×640 frame size, 30× objective, ×6.0 zoom, and 0.74 µm z-stack interval for quantification in P14 muscles; and 800×800 frame size, 30× objective, ×4.5 zoom, and 0.74 µm z-stack interval for quantification in P28 muscles. For quantification of Gl muscles, the number of NMJs was quantified along the entire Gl nerve within the Gl muscles in P0 and P14. Quantification of the length of secondary branches, end plate area occupied by motor axons or AChR clusters in P0, and the nerve terminal area, AChR cluster area, compactness, and fragmentation in P14 and P28 was performed using ImageJ software (*Jones et al., 2016*). The number of axons extending the border of the target muscles was calculated by quantifying the overshooting axons per 100 µm of end plate. The proportion of polyinnervated, faint, denervated NMJs was quantified within three to four random fields per animal. The proportion of polyinnervated NMJs was quantified in Gl nerves adjacent to Tfl muscles. The number of axonal inputs was quantified in individual NMJs in P14. In Tfl muscles, we quantified the number of NMJs along the entire nerve within the Tfl muscles at P14, and the number of axonal inputs in individual NMJs at P28.

## Skeletal staining and X-ray imaging

E17.5 mouse embryos were prepared by removing the skin, viscera, and adipose tissues and fixed in 95% ethanol for at least 5 days. The tissues were incubated in acetone for 2 days to remove fat and stained in staining solution (1 volume of 0.3% Alcian blue in 70% ethanol, 1 volume of Alizarin red in 95% ethanol, 1 volume of glacial acetic acid, and 17 volumes of 70% ethanol) for 3 days at

37°C. Stained tissues were rinsed in distilled water and cleared in 1% KOH, 20% glycerol in 1% KOH, 50% glycerol in 1% KOH, and 80% glycerol in 1% KOH, sequentially, for several days each. Cleared embryos were stored in 100% glycerol before imaging.

## Whole-mount immunostaining for axon tracing

E11.5, E12.5, and E13.5 *Hb9::GFP* embryos were dissected, and samples were immersed in 4% PFA and washed in PBS overnight. Tissues were cleared using a three-dimensional imaging kit (Binaree). Images were captured using a Zeiss confocal microscope using the ZEN software (Zeiss).

## RNA-seq and bioinformatics analysis

For the RNA-seq analysis of ventral brachial and lumbar spinal cords, E12.5 embryonic spinal cords (three heterozygote animals and three homozygote animals) were harvested and trimmed around the *Hb9::GFP* reporter in ice-cold RNase-free PBS. Total RNA was extracted using NucleoSpin RNA XS, Micro kit (MACHEREY-NAGEL) for RNA purification, and RNA quality and quantity were assessed using a 2100 Bioanalyzer System (Agilent). All samples showed high-quality scores between 9.6 and 10 RIN. 500 ng of total RNA per sample was used for the construction of sequencing libraries, which was amplified using the TruSeq Stranded Total RNA LT Sample Prep Kit (Gold). Library preparation and bulk RNA-seq were performed by Macrogen (Seoul, Korea) using the manufacturer's reagents and protocols. Libraries were sequenced on an illumine NovaSeq platform. Raw sequencing files were assessed by FastQC (v0.11.7) for quality, and sequence reads were trimmed for adaptor sequence and low-quality sequence by Trimmomatic (v0.38). The processed reads were mapped to the mm10 mouse reference genome using HISAT2 (v2.1.0) and Bowtie2 (v2.3.4.1) for further processing. Assembly of aligned reads and abundance estimations were performed using StringTie (v2.1.3b), and fragments per kilobase of transcript per million fragments mapped values and transcript per million values were acquired. Analysis of DEGs was conducted using DESeq2.

## GO analysis

GO analysis based on bulk RNA-seq data was performed using Metascape (http://metascape.org) (*Zhou et al., 2019*).

## Statistical representation in figures

Error bars represent SEM. Box plots show 25th percentile, median, and 75th percentile values, with whiskers indicating 10th and 90th percentile values. Outlying values are shown as symbols. Statistical analyses were performed with GraphPad Prism 9 (GraphPad Software, San Diego, CA, USA).

## Use of published datasets

For the evaluation and comparison of candidate genes downregulated in *Isl2*-deleted embryos, we compared our gene lists with a previously published dataset (ArrayExpress accession: E-MTAB-10571) (*Amin et al., 2021*). The raw data from the study by *Amin et al., 2021*. were obtained from Array-Express accession: E-MTAB-10571. Reads from wild-type samples (E12 *Hb9::GFP*) were mapped to the mm10 mouse reference genome, and raw unique molecular identifiers were produced using Cell Ranger (version 3.1.0, 10x Genomics). The expression matrices were processed in R (v.3.6.0) using Seurat (v.3.2.3). Cells with fewer than 1000 detected transcripts were excluded and cells with over 8% of mitochondrial gene read counts were excluded. Filtered gene-barcode matrices were normalized using the *NormalizeData* function with *LogNormalize*, and the top 2000 variable genes were identified using the 'vst' method in *FindVariableFeatures*. Gene expression matrices were scaled using the *ScaleData* function. Principal component analysis (PCA) and UMAP were performed using the 20 principal components. To remove the batch effect, *RunFastMNN* was applied to the PCA matrix with SampleID as the batch key. Dimensionality reduction was performed separately for all cells and the interneuron-excluded subset. Cells were partitioned into 31 clusters with a resolution parameter of 2. Neuron subtype identification was performed as previously described (*Amin et al., 2021*). MN subtypes were identified according to the expression of known marker genes. The number of cells in each neuron subtype was as follows: 19 cells in pMNs, 672 cells in immature MNs, 3464 cells in postmitotic MNs, and 1135 cells in interneurons.

Further downstream analyses within LMC clusters were performed using Seurat v.4.0.2 R packages. To analyze motor pools at the L1 to L4 levels, we isolated LMC subclusters that expressed *Hoxd9*, *Hoxc10*, *Hoxd10*, *Hoxa11*, *Hoxc11*, and *Hoxd11*. As a result, LMC clusters were further divided into seven clades composed of 276 cells: sb.LMCl.1.v, sb.LMCl.1.ta, sb.LMCl.2, sb.LMCm.1, sb.LMCm.2, and sb.LMCm.3. *Lhx1*, an LMCl marker, was enriched in sb.LMCl.1.v, sbLMCl.1.ta, and sb.LMCl.2 clades, whereas *Isl1*, an LMCm marker, was enriched in LMCm.1, LMCm.2, and LMCm.3. Visualized data were plotted using *VinPlot*, *DotPlot*, *DoHeatmap*, and *FeaturePlot* functions from the Seurat R packages.

## Acknowledgements

This work was supported by grants from NRF (NRF-2018R1A5A1024261, NRF-2022M3E5E8081194, NRF-2023R1A2C1002690), SNUH Lee Kun-hee Child Cancer & Rare Disease Project (22B-001-0500) and a GIST Research Institute (GRI) IIBR funded by the GIST in 2023. We thank Drs. Samuel Pfaff, Jeremy Dasen, and Bennett Novitch for their generous contribution of the research materials used in the study. We thank Jung Eun Kim and Won-Young Lee for their support in experimental setup, Hwan Kim (GIST Central Research Facility) for technical assistance in microscopy, and Myungin Baek for critical comments on the manuscript.

## Additional information

### Funding

| Funder | Grant reference number | Author |
|---|---|---|
| National Research Foundation of Korea | NRF-2018R1A5A1024261 | Mi-Ryoung Song |
| National Research Foundation of Korea | NRF-2022M3E5E8081194 | Mi-Ryoung Song |
| National Research Foundation of Korea | NRF-2023R1A2C1002690 | Mi-Ryoung Song |
| Gwangju Institute of Science and Technology | the GIST Research Institute (GRI) IIBR in 2023 | Mi-Ryoung Song |
| SNUH Lee Kun-hee Child Cancer & Rare Disease Project | 22B-001-0500 | Mi-Ryoung Song |

The funders had no role in study design, data collection and interpretation, or the decision to submit the work for publication.

### Author contributions

Yunjeong Lee, Jihwan Park, Data curation, Formal analysis, Validation, Investigation; In Seo Yeo, Data curation, Investigation, Methodology; Namhee Kim, Data curation, Formal analysis, Validation, Investigation, Methodology; Dong-Keun Lee, Kyung-Tai Kim, Data curation, Validation, Investigation; Jiyoung Yoon, Data curation, Formal analysis, Investigation; Jawoon Yi, Formal analysis; Young Bin Hong, Data curation, Validation, Investigation, Methodology; Byung-Ok Choi, Resources, Supervision; Yoichi Kosodo, Daesoo Kim, Methodology; Mi-Ryoung Song, Conceptualization, Supervision, Funding acquisition, Writing - original draft

### Author ORCIDs

Yunjeong Lee ⓘ http://orcid.org/0009-0002-6694-9991
Jihwan Park ⓘ http://orcid.org/0000-0002-5728-912X
Mi-Ryoung Song ⓘ http://orcid.org/0000-0003-0350-0863

### Ethics

This study was performed in strict accordance with the recommendations in the Guide for the Care and Use of Laboratory Animals of Gwangju Institute of Science and Technology (GIST, No.

GIST2021-072). All mice were housed and cared for in an Association for Assessment and Accreditation of Laboratory Animal Care (AAALAC)-accredited animal facility under specific pathogen-free conditions.

## Decision letter and Author response
Decision letter https://doi.org/10.7554/eLife.84596.sa1
Author response https://doi.org/10.7554/eLife.84596.sa2

---

## Additional files

### Supplementary files
• Supplementary file 1. Genotype distribution of Isl2 knockout (KO) and conditional KO mice. Genotype distribution of *Isl2*-null and conditional KO mice. Expected Mendelian inheritance and observed inheritance after breeding between *Isl2* heterozygotes or between *Isl2 +/-; Olig2^Cre* and *Isl2^F/F*. p-Values of the Mendelian ratios were calculated using chi-square tests.

• Supplementary file 2. Experimental sample sizes. Experimental sample sizes for figures and figure supplements.

• Supplementary file 3. List of differentially expressed genes (DEGs) downregulated in *Isl2*-null brachial and lumbar spinal cords. (a) List of DEGs downregulated in *Isl2*-null brachial spinal cords. (b) List of DEGs downregulated in *Isl2*-null lumbar spinal cords. (c) List of top 30 enriched genes in sb.LMCl.2 subcluster.

• MDAR checklist

### Data availability
Sequencing data have been deposited in GEO under accession codes GSE217297.

The following dataset was generated:

| Author(s) | Year | Dataset title | Dataset URL | Database and Identifier |
|---|---|---|---|---|
| Song MR, Lee Y | 2022 | Transcriptional control of motor pool formation and motor circuit connectivity by the LIM-HD protein Isl2 | https://www.ncbi.nlm.nih.gov/geo/query/acc.cgi?acc=GSE217297 | NCBI Gene Expression Omnibus, GSE217297 |

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
