## [Editor Report]

This paper will be of interest to developmental biologists who study the gene regulatory mechanisms necessary for neuronal identity and circuit assembly. The study presents important findings regarding the role of the LIM homeodomain transcription factor Isl2 in the development of spinal motor neurons. While the importance of Isl2 for the acquisition of axial and visceral motor neuron development was already described in the literature, the data convincingly describe an additional role in the differentiation of a subset of limb-innervating motor neurons.

---

## [Decision Letter]

**Decision letter after peer review:**

Thank you for submitting your article "Transcriptional control of motor pool formation and motor circuit connectivity by the LIM-HD protein Isl2" for consideration by *eLife*. Your article has been reviewed by 3 peer reviewers, and the evaluation has been overseen by a Reviewing Editor and Marianne Bronner as the Senior Editor. The following individuals involved in the review of your submission have agreed to reveal their identity: Paschalis Kratsios (Reviewer #1); Niccolò Zampieri (Reviewer #2).

Essential revisions:

1) The issue of cell autonomy was raised by all three reviewers because the study primarily focuses on global Isl2 KO mice. Additional characterization of the conditional Olig2Cre;Isl2 mice would significantly strengthen the conclusions. The authors do not need to repeat every experiment in the conditional Olig2Cre;Isl2 mice, but the reviewers strongly encourage the authors to conduct the same behavioral and connectivity experiments in the conditional mice.

2) The writing is at times not very clear and makes it difficult to appreciate the findings and their relevance in the context of the literature. Please address specific comments by reviewer #1 (presentation of data on Figure 2; grammar issues and typos) and reviewer #2 (the Introduction is difficult to read; more emphasis should be given to what is known about Isl2 and Pea3 mutant mice, which is very relevant for the paper).

3) Based on this study and previous studies (e.g., Catela et al. 2016, Cell Reports, PMID: 26904955), it seems that there is differential regulation of Pea3 at brachial and lumbar levels. The Pea3 phenotypes described by Arber et al. were very dramatic-almost complete lack of innervation. Here the phenotypes appear to be less so and mostly at the NMJ, suggesting Pea3 partially controls aspects of the lumbar motor pool properties. So, if Isl2 regulates Pea3 differentially, it's possible that Pea3 is not gdnf-dependent at lumbar levels (is this known?). The explant experiments aren't that convincing because they contain all lumbar motor axons-if only a few pools are regulated by Isl2 why do none of them respond to gdnf. The authors should have a more thoughtful discussion about the gene regulatory networks at brachial vs lumbar levels and what the significance might be. They also must cite Catela et al. 2016, which nicely showed that Ret expression is dependent on brachial Hox genes, and that Hoxc10/d10 can also induce Ret expression at lumbar levels. They should incorporate these findings in their discussion.

4) Improve quantifications: The way MN numbers are given throughout the manuscript is confusing. The data should be recalculated and presented as average MN numbers/mouse (n=1 mouse and not 1 section). In Figure 2, quantification would be really good to have at both stages. The statistical difference in MMC and PGC numbers (Figure 2f) observed between WT and KO mice should be briefly commented on, in light of Thaler's results and in the context of the ectopic motor neurons.

5) Figure 3: The markers used in Figure 3a do not allow to clearly distinguish MMC, LMCm and LMCl neurons. A different combination should be used such as Lhx3 (MMC) Isl1 (LMCm) and Hb9 (LMCl). In addition, spatial analysis of columnar and divisional segregation should be provided to better support the statement.

6) Further experimentation and/or discussion is needed to clarify which exact motor pools are affected in Isl2 mutant mice. See comments by reviewers 1 and 3.

*Reviewer #1 (Recommendations for the authors):*

Figure 2 is confusing. The presentation and discussion of the data in Results (lines 149-162) need to be improved. The authors should also present their data more clearly in the context of Thaler et al. (2004), as that study found increased numbers of somatic motor neurons and decreased numbers of visceral motor neurons in the thoracic spinal cord. By doing so, the novelty of the current study will be clearer. Perhaps, switching the order of Figures 2 and 3 would address this issue and improve the flow.

In the transcriptomic analysis, the authors do a remarkable job to identify motor neuron-expressed genes and define subclusters of LMC motor pools. Although the RNA-Seq is done in bulk and without motor neuron specificity, the authors leverage published RNA-Seq data to ascribe MN specificity to their dataset. Hence, they identified several putative targets, which they validate with ISH (Figure 6H). However, they are strongly encouraged to also validate with ISH putative markers of the subcluster LMCl.2 in Isl2 global or conditional mice. This experiment would significantly strengthen the authors' conclusions (lines 356-366) about which motor pools are affected by Isl2 loss, especially because Pea3 is a marker for multiple motor pools.

The study will be strengthened by conducting hindlimb locomotion analysis in Olig2Cre;Isl2 fl/fl mice.

The text can be significantly improved by eliminating typos and improving grammar. The authors should go carefully through the text, sentence by sentence, and make the necessary changes.

In the first section of Results, the authors can cite Thaler et al. (2004), as their results are consistent with that study.

*Reviewer #2 (Recommendations for the authors):*

In my opinion, the writing is at times not very clear and makes it difficult to appreciate the findings and their relevance in the context of the literature. The experiments are well executed and appropriately analyzed, there are a few things that can be improved, more to it in the following detailed comments:

1) I find the introduction difficult to read, it is more like a list of facts than a coherent narrative to properly frame the work. For example, the bit about neuromesodermal progenitors (lines 66-71), while interesting does not fit with the rest of the story. Also, at times the balance between subjects seems a bit off, more emphasis should be given to what is known about Isl2 and Pea3 mutant mice, which is very relevant to the paper.

2) Line 150: it is not specified which Isl2 KO mice is used. At the very least there should be a reference (Thaler et al., 2004). However, here it is very important to comment that these are the same mice that have been used in a previous study characterizing the role of Isl2 in spinal motor neuron development. In addition, the mice were originally reported to "die within 24 hr of birth" (Thaler et al., 2004), but the authors found that "approximately 10% survived" (lines 315-317). This important information should be introduced at the beginning when the mouse line is first introduced. It will be immediately relevant as in line 160 adult isl2 KO mice are used.

3) Line 159: "Mislocalized motor neurons remained in the adult Isl2-null lumbar spinal cord with a similar number of motor neurons" Mislocalised motor neurons are not very evident from pictures provided in figure 2h higher magnifications would be helpful. Regarding the "similar number", I could not find quantification of mislocalised motor neurons either in embryos or adults, thus a quantitative statement is not justified. Indeed, quantification would be really good to have at both stages.

4) The statistical difference in MMC and PGC numbers (Figure 2f) observed between WT and KO mice should be briefly commented on, in light of Thaler's results and in the context of the ectopic motor neurons.

5) Line 169: "We similarly found that MMC neurons were scattered dorsolaterally and located ectopically, either within the LMC columns or immediately dorsal to the motor columns in Isl2-deficient (Figure 3a)" The markers used in Figure 3a do not allow to clearly distinguish MMC, LMCm and LMCl neurons. A different combination should be used such as Lhx3 (MMC) Isl1 (LMCm) and Hb9 (LMCl). In addition, spatial analysis of columnar and divisional segregation should be provided to better support the statement.

6) Figure 4b: I believe there must be a mistake as it is clear from the pictures in 4a that there are very few Pea3+ neurons in Isl2 KO mice, however in the plots in 4b (last row the three plots on the right) there are huge red contours for Pea3 neurons more or less similar to control mice. Indeed, the difference is clearly evident in the dot plots in Figure 4 supp 1b.

7) It is great that the authors generated an Isl2 conditional allele, however, its phenotype in combination with olig2::cre is not described other than in the experiments in Figure 5 aimed at studying the downregulation of Pae3. Does the Isl2 cKO mouse recapitulate the overall defects observed in Isl2 KO in terms of motor neuron generation, spatial organisation, connectivity, and behaviour? Comparison of the two can be very informative especially regarding the behavioural and sensorimotor connectivity phenotypes as many other cells (i.e., all somatosensory neurons) express Isl2 during development.

8) Do you observe arborization, sensorimotor connectivity, and NMJ defects shown for Gl motor neurons also in Rf and Tfl motor pools? It seems like it would be the case but it would be a great addition in order to generalise the role of Isl2-Pea3 transcriptional axis on all pools expressing it.

*Reviewer #3 (Recommendations for the authors):*

Specific concerns:

1) The behavioral phenotypes seen in the videos appear to be very dramatic, especially in the hindlimbs. However, the authors show that most hindlimb muscles are properly innervated, with the exception of the Gl. For example, the Rf seems to be properly innervated but its weight is dramatically reduced in Figure 9. What is the cause for this? Do the authors believe that the behavior correlates to defects in Gl development and connectivity or are there non-MN contributions to these phenotypes from sensory neurons and interneurons?

2) The connection between Isl2/gdnf signaling and Pea3 expression is not experimentally supported. The authors do show a specific downregulation of Pea3 at lumbar levels of the spinal cord but whether this is mediated through gdnf is not established. Is Ret or any other component of this signaling pathway downregulated in the RNA-seq data? The fact that lumbar explants do not respond to gdnf may suggest this but the connection is not explicitly shown.

3) The loss of synapses in Fig7b is interpreted as Isl2 controlling Gl connectivity. Since this is not a conditional allele, the synaptic loss could be due to defects in sensory neurons. The authors also show an ectopic soma (although this is not really well defined). Is there a difference between ectopic and Gl MNs that settle in the correct position? Are the authors suggesting that cell body position influences connectivity in a way that ectopic MNs lose more connections? This is not quantified or discussed.

4) The way MN numbers are given throughout the manuscript is confusing. The authors quantify MN/section. However, especially for motor pool counts, this results in a lot of variability since at the beginning and end of each motor pool you have 1-2 MNs, while higher numbers are in the middle. The data should be recalculated and presented as average MN numbers/mouse (n=1 mouse and not 1 section). This would reduce variability within the datasets.

5). Figure 1: The authors discuss the different spatiotemporal expressions of Isl1 and Isl2 but this is not shown (maybe inferred from RNAseq). Only one time point is shown in the figure (e12.5). It would be nice to show the temporal expression of both TFs at different embryonic time points and also in combination with motor pool markers. For example, the overlap of Isl2 with Pea3 at brachial and lumbar levels is not shown. Showing which motor pools express high and sustained levels of Isl2 in this figure would be a nice setup for the rest of the story.

6). Figure 2-4: MN numbers should be recalculated and presented so that n=1 mouse. It is likely that this might change some of the data. For example, in Figure 3a, there is a visible reduction of Scip+ MNs. When the source data for Figure 3c was reanalyzed, there seems to be a 30% reduction of FCU neurons with a P=0.03 (not significant but close, perhaps would be significant if n>3). Is Isl2 highly expressed in Scip+ MNs at brachial levels (see point for Figure 1). Also the MN positional changes should be quantified in these figures.

7). Figure 6: After comparing their RNAseq data to scRNAseq the authors mention that only 50% seem to be MN-specific. Is this technical (not deep enough sequencing in the scRNA-seq datasets) or are the rest of the genes in interneurons? This relates to the comment about other affected populations in the null mouse. In the in situ data, the ventrolateral pools are indicated, however, the changes appear to be broader, affecting the entire lumbar spinal cord. The nNos data should be shown separately since that is in the thoracic spinal cord.

8). Figure 7: In the text, it is mentioned that the Pea3 pools are Gl, Rf, and Tfl and that a retrograde tracer was injected into individual muscles. However, only the Gl data are shown. What about the other pools? Do they have normal dendrites? It seems that Pea3 is not the whole story as Rf has normal NMJ innervation. What about sensory innervation? Is that synapse loss specific to Gl MNs? What about Rf and Tfl? What about non-Pea3+ pools?

---

## [Author Response]

Essential revisions:1) The issue of cell autonomy was raised by all three reviewers because the study primarily focuses on global Isl2 KO mice. Additional characterization of the conditional Olig2Cre;Isl2 mice would significantly strengthen the conclusions. The authors do not need to repeat every experiment in the conditional Olig2Cre;Isl2 mice, but the reviewers strongly encourage the authors to conduct the same behavioral and connectivity experiments in the conditional mice.

Initially, our experiments primarily involved the use of global *Isl2* KO mice, and more recently, we generated *Isl2* conditional mice to address the issue of cell autonomy. Through meticulous observation, we identified that about 10% of global *Isl2* KO survived, allowing us to effectively examine their postnatal phenotypes. Regrettably, the *Isl2* conditional KO (cKO) mice showed a different survival rate, as they all succumbed within a day after birth. Thus, our ability to perform experiments utilizing postnatal animals was limited.

Detailed results and the interpretation surrounding the observed mortality discrepancy in these two mouse lines have been comprehensively presented in Table S1, Supplementary Videos 1-3, and elaborated within the Results and Discussion section.

Page 10: “A previous study by Thaler et al. (2004) had established that *Isl2* KO mice were unable to survive beyond a day after birth (Thaler et al., 2004). Intriguingly, our study revealed that *Isl2* cKO mice also died on the first day after birth, while 11% of *Isl2* KO mice managed to survive (Supplementary Table S1, Supplementary Video S3). While the precise underlying cause for this disparity in survival rates among *Isl2* KO mutant mouse lines requires further investigation, we were able to examine adult motor neurons in the absence of Isl2.”

Page 16: “Thaler et al. initially reported that *Isl2* KO mice died within a day after birth due to respiratory failure as well as motor neuron developmental defects (Thaler et al., 2004). In contrast, we found that the identical mouse line exhibited a greater survival rate, about 10%. On the other hand, motor neuron-specific *Isl2* conditional KO mice showed similar lethality, with all pups dying on the first day after birth. There are several possibilities to explain the variable mortality among different mouse lines lacking *Isl2*. First, acute elimination of *Isl2* gene in developing motor neurons could be more detrimental, minimizing potential developmental compensation through altered expression of other genes reconciling the impact of the target knockout. Second, the genetic background of the mice may play a role, as it has been reported that different genetic backgrounds can cause varying phenotypes and lethality in knockout mouse models (El-Brolosy and Stainier, 2017). For instance, the lethality of Lhx4 embryos was found to be different in C57BL/6J relative to the mixed background (Gergics, Brinkmeier, and Camper, 2015). Similarly, the genetic background of original *Isl2* global null mice could be different from ours, with these mice being occasionally backcrossed onto the C57BL/6 genetic background. Despite the variable survival rates, the defects found in *Isl2* null and conditional KO embryos were essentially identical, including scattered lumbar motor pools, absence of Etv4 expression, and defective NMJs, all of which are closely related to the adult phenotypes found in global *Isl2* KO mice. These findings further highlight the critical role of Isl2 in motor neuron development and locomotion. Overall, our study provides a comprehensive analysis of the broad spectrum of phenotypes resulting from removal of *Isl2* expression.”

Nevertheless, we successfully investigated the progress of NMJ development in *Isl2* cKO at E18.5, just prior to birth, and identified congruent developmental anomalies comparable to *Isl2* null mice. Consequently, the phenotypes we observed during the embryonic stage were identical in both mouse lines, i.e., dispersed motor neurons, Etv4 expression absence in selective lumbar motor pools, disrupted NMJ formation within those pools. The new obtained data, along with pertinent explanations, have been integrated into Figure 8—figure supplement 2 and incorporated within the Results and Discussion section, as outlined below.

Page 12: “Due to the postnatal lethality of *Isl2* cKO, we analyzed the muscles of these mice at E18.5, prior to birth, when NMJ formation is in progress (Figure 8—figure supplement 2). GFP-labeled axons arrived in *Isl2-*deficient Gl muscles; however, their terminal arbors were defasciculated and misrouted. Fluorescence intensity of α-bungarotoxin (BTX) at the NMJs was greatly reduced in *Isl2* cKO. Furthermore, AChR clustering appeared to be incomplete, showing slender and elongated NMJs, or some NMJs were abnormally concentrated in proximity to the primary axon shaft in *Isl2* cKO (Figure 8—figure supplement 2b).”

2) The writing is at times not very clear and makes it difficult to appreciate the findings and their relevance in the context of the literature. Please address specific comments by reviewer #1 (presentation of data on Figure 2; grammar issues and typos) and reviewer #2 (the Introduction is difficult to read; more emphasis should be given to what is known about Isl2 and Pea3 mutant mice, which is very relevant for the paper).

We have thoroughly revised the Introduction, Result and Discussion section of our manuscript, aiming to offer a more comprehensive and coherent analysis of our results and their correlation with prior research in the field. Furthermore, the manuscript has undergone rigorous editorial refinement by a professional editor.

3) Based on this study and previous studies (e.g., Catela et al. 2016, Cell Reports, PMID: 26904955), it seems that there is differential regulation of Pea3 at brachial and lumbar levels. The Pea3 phenotypes described by Arber et al. were very dramatic-almost complete lack of innervation. Here the phenotypes appear to be less so and mostly at the NMJ, suggesting Pea3 partially controls aspects of the lumbar motor pool properties. So, if Isl2 regulates Pea3 differentially, it's possible that Pea3 is not gdnf-dependent at lumbar levels (is this known?). The explant experiments aren't that convincing because they contain all lumbar motor axons-if only a few pools are regulated by Isl2 why do none of them respond to gdnf. The authors should have a more thoughtful discussion about the gene regulatory networks at brachial vs lumbar levels and what the significance might be. They also must cite Catela et al. 2016, which nicely showed that Ret expression is dependent on brachial Hox genes, and that Hoxc10/d10 can also induce Ret expression at lumbar levels. They should incorporate these findings in their discussion.

We appreciate the Reviewer’s insightful comments. As suggested, we have cited the work of Catela et al. and improved the text with a more comprehensive description and discussion, as presented below.

Page 15: “Additional compelling evidence that Isl2 mainly functions in Etv4^+^ motor pools comes from the striking similarities and differences observed in the mutant phenotypes of Isl2 and the GDNF-Etv4 pathway, requiring a detailed comparison. GDNF has been recognized as a critical factor for the survival and innervation of motor neurons during early motor neuron development (Henderson et al., 1994). Deletion of *GDNF* or its receptor *Ret* or *Gfra1* leads to impaired innervation of the peroneal nerve, resulting in no or disrupted axon innervation in major hindlimb muscles, such as Edl, Atib, psoas major, Gmax, and Tfl, to varying degrees (Bonanomi et al., 2012; Gould, Yonemura, Oppenheim, Ohmori, and Enomoto, 2008; Kramer et al., 2006). In contrast, *Isl2* mutant mice showed normal axon projection of the peroneal nerve and relatively normal axon innervation in most hindlimb muscles, unlike motor axons of mice deficient in *GDNF* or its receptors. Notably, discrepancies between the roles of GDNF and Isl2 also emerged in types of motor pools affected. The removal of *GDNF*, its receptor *Ret* or *Gfra1,* or *Etv4* has been mostly characterized in CM and LD motor pools, the Etv4-expressing motor pools at the forelimb level, but their roles in the lumbar motor pools have not been addressed (Arber, Ladle, Lin, Frank, and Jessell, 2000; Haase et al., 2002; Livet et al., 2002; Vrieseling and Arber, 2006). In our study, we observed a selective loss of *Etv4* expression only in the lumbar spinal cord of *Isl2* KO mice, specifically in Gl and Tfl motor pools. These motor pools displayed a phenotype almost identical to CM and LD motor pools of *Etv4*-null mice, including mispositioned cell bodies, impaired dendrite and axon arborization, defective NMJs, and reduced sensorimotor connectivity. In addition, acute knockdown of *Isl2* in the chicken neural tube and in motor neuron-specific *Isl2* cKO mice was sufficient to abolish *Etv4* transcripts, suggesting that Isl2 positively regulates *Etv4* expression in a cell-autonomous manner. Thus, GDNF likely plays broader roles in guiding most LMC projections and also promoting differentiation of Etv4-expressing motor pools at both limb levels. In contrast, Isl2 plays a precise role in fine-tuning the position of motor neurons, including MMC and LMCl, and also functions in selected lumbar motor pools at the hindlimb, particularly in establishing sensorimotor connectivity during the late period of development.

Nevertheless, Etv4 expression in the CM motor pool at the brachial level was intact in both *Isl2* KO mice and cKO mice, implying that another complex genetic program works in conjunction with Isl2 to contribute to segment-specific regulation. For instance, multiple *Hox* genes control GDNF signaling, with *Hoxc6* activating *Ret* and *Gfra1* expression in rostral brachial spinal cords, regulating terminal arborization of CM and LD motor pools, while *Hoxc8* activates *Ret* and *Gfra3* expression in caudal brachial spinal cords, controlling limb innervation of distal nerves (Catela, Chen, Weng, Wen, and Kratsios, 2022). Similar regulatory pathways involving Hox/Mes in *Ret* expression have been suggested at lumbar levels, with misexpression of *Hoxc10* or *Hoxd10* in chicken embryos resulting in *Ret* expression arising in the neural tube (Catela et al., 2022). Our bulk RNA-seq analysis identified *Hoxc11* as one of the DEGs, indicating a potential compromise of proper lumbar identity in the absence of Isl2. However, the expression of *Ret* and *Gfra1* was not changed in *Isl2* KO mice when assessed by bulk RNA-seq analysis and in situ hybridization, and overall hindlimb motor projections remained intact. Although further investigation is necessary to fully elucidate the hierarchy among Hox, GDNF and Isl2, it is evident that each factor plays a crucial role, to varying degrees, in the proper expression of *Etv4* within proximal hindlimb motor pools.”

4) Improve quantifications: The way MN numbers are given throughout the manuscript is confusing. The data should be recalculated and presented as average MN numbers/mouse (n=1 mouse and not 1 section). In Figure 2, quantification would be really good to have at both stages. The statistical difference in MMC and PGC numbers (Figure 2f) observed between WT and KO mice should be briefly commented on, in light of Thaler's results and in the context of the ectopic motor neurons.

We re-analyzed the motor neuron count, presenting it as an average number of motor neurons per section derived from individual embryos of each genotype across Figure 2 to Figure 7. We also provide a succinct comparison of our quantification outcomes with those previously reported by Thaler et al.

Page 6: “Our findings align with previous results, including a reduced number of visceral motor neurons, reduced nNOS expression, and abnormal dorsomedial migration of motor neurons, reported by Thaler et al. Our study additionally reveals the appearance of ectopic motor neurons at the limb levels, primarily consisting of MMC and LMCl cells.

We have included cell count of ectopic motor neurons in embryos in Figure 2—figure supplement 1c. The quantification of adult motor neurons has been relocated to Figure 7 to improve the continuity. Since the columnar identity is less discernible in adult motor neurons, our quantification focused solely on determining the total motor neuron count within each segment (Figure 7b). Furthermore, our investigation of the medio-lateral distribution of ectopic motor neurons has revealed the presence of such neurons in *Isl2* null mice (Figure 7c).”

Page 11: “Overall, the total number of ChAT^+^ motor neurons remained unchanged, and mislocalized motor neurons were observed in the P21 *Isl2-*null lumbar spinal cord (Figure 7a-c).”

5) Figure 3: The markers used in Figure 3a do not allow to clearly distinguish MMC, LMCm and LMCl neurons. A different combination should be used such as Lhx3 (MMC) Isl1 (LMCm) and Hb9 (LMCl). In addition, spatial analysis of columnar and divisional segregation should be provided to better support the statement.

To better distinguish motor columns, we used additional markers Hb9, Lhx3, Lhx1 for our analysis in Figure 3 as the Reviewer suggested. sMN (Hb9), MMC (Hb9, Lhx3), HMC (Hb9, Scip), LMCm (Hb9low, Isl1, Foxp1), LMCl (Hb9high, Lhx1, Foxp1) were carefully defined using these markers. Further, FCU (Scip), CM (Etv4, Isl1), LD (Etv4, Lhx1) motor pools were defined as well. Contour, density and spatial plots of motor columns were provided in Figure 3-supplement 2 and 3.

6) Further experimentation and/or discussion is needed to clarify which exact motor pools are affected in Isl2 mutant mice. See comments by reviewers 1 and 3.

We sincerely appreciate the valuable feedback offered by the Reviewer. Following the suggestion of our Reviewer, we expanded our study to encompass the developmental examination of other Etv4-expressing motor pools, namely Rf and Tfl, in addition to Gl. Our comprehensive analysis uncovered noteworthy developmental impacts on the Tfl motor pools, while the Rf motor pools remained unaffected. Considering the divergence of Tfl axons from Gl projections, it is plausible that these motor pools share analogous developmental genetic programs. A previous study reported that Etv4 expression is transient in Rf motor pools, lasting until E13.5 (De Marco Garcia and Jessell, 2008). Thus, the postnatal development of Rf motor pools could be independent of Etv4.These results support the notion that Isl2 exerts both a universal and distinct influence on Etv4+ motor pools, albeit to varying extents. We have supplemented our findings with illustrative images and quantification data (Figure 7d, Figure 8h-j and Figure 8—figure supplement 3), providing further elaboration within the Results and Discussion sections of our manuscript.

Page 11: “Similarly, Tfl motor pools, whose axons diverged from one of the major branches of Gl ones, also displayed shorter and lesser dendritic arbors in *Isl2* KO mice (Gould et al., 2008). The position and dendritic arborization of Rf motor pools remained unchanged in *Isl2* KO mice (Figure 7d). Considering that Etv4 expression has been previously reported to be transient until E13.5 in Rf motor pools, it may not play a significant role in the dendritic arborization of Rf motor pools (De Marco Garcia and Jessell, 2008).”

Page 13: “In Tfl muscle, abnormal AChRs were formed along axon branches at P0, and a drastic loss in AChR clusters and polyinnervation were observed at P28 in *Isl2* KO mice (Figure 8h, i and j). However, Rf muscles, whose motor pools transiently express Etv4, showed normal arrangement of NMJs (Figure 8—figure supplement 3) (De Marco Garcia and Jessell, 2008).”

Page 15: “The removal of *GDNF*, its receptor *Ret* or *Gfra1,* or *Etv4* has been mostly characterized in CM and LD motor pools, the Etv4-expressing motor pools at the forelimb level, but their roles in the lumbar motor pools have not been addressed (Arber et al., 2000; Haase et al., 2002; Livet et al., 2002; Vrieseling and Arber, 2006). In our study, we observed a selective loss of *Etv4* expression only in the lumbar spinal cord of *Isl2* KO mice, specifically in Gl and Tfl motor pools. These motor pools displayed a phenotype almost identical to CM and LD motor pools of *Etv4*-null mice, including mispositioned cell bodies, impaired dendrite and axon arborization, defective NMJs, and reduced sensorimotor connectivity. In addition, acute knockdown of *Isl2* in the chicken neural tube and in motor neuron-specific *Isl2* cKO mice was sufficient to abolish *Etv4* transcripts, suggesting that Isl2 positively regulates *Etv4* expression in a cell-autonomous manner. Thus, GDNF likely plays broader roles in guiding most LMC projections and also promoting differentiation of Etv4-expressing motor pools at both limb levels. In contrast, Isl2 plays a precise role in fine-tuning the position of motor neurons, including MMC and LMCl, and also functions in selected lumbar motor pools at the hindlimb, particularly in establishing sensorimotor connectivity during the late period of development.”

Reviewer #1 (Recommendations for the authors):Figure 2 is confusing. The presentation and discussion of the data in Results (lines 149-162) need to be improved. The authors should also present their data more clearly in the context of Thaler et al. (2004), as that study found increased numbers of somatic motor neurons and decreased numbers of visceral motor neurons in the thoracic spinal cord. By doing so, the novelty of the current study will be clearer. Perhaps, switching the order of Figures 2 and 3 would address this issue and improve the flow.

Thank you for the valuable feedback from the Reviewer. We re-analyzed number of motor neurons as an average number of motor neurons per section derived from one embryo of each genotype in Figure 2 to Figure 7. Thaler et al. presented Hb9+ motor neuron counts across spinal cord segment in a histogram, showing a slightly greater count in *Isl2* null mice. However, they did not conduct statistical analysis on this. In contrast, our study separately quantified MMC and HMC, averaging their numbers along the thoracic segment. According to these criteria, the number of MMC and HMC in *Isl2* KO mice did not significantly differ. We also briefly discuss about our quantification results compared to the previous one done by Thaler et al. as follows.

Page 6: “Our findings align with previous results, including a reduced number of visceral motor neurons, reduced nNOS expression, and abnormal dorsomedial migration of motor neurons, reported by Thaler et al. Our study also reveals the appearance of ectopic motor neurons at the limb levels, primarily consisting of MMC and LMCl cells.”

As Figure 3 and 4 collectively present the motor column distribution, we have chosen to retain the initial sequence of Figure 2 and 3. In this revised arrangement, the outcomes pertaining to adult animals (Figure 2e) have been shifted to Figure 7a-c to ensure a smoother transition.

In the transcriptomic analysis, the authors do a remarkable job to identify motor neuron-expressed genes and define subclusters of LMC motor pools. Although the RNA-Seq is done in bulk and without motor neuron specificity, the authors leverage published RNA-Seq data to ascribe MN specificity to their dataset. Hence, they identified several putative targets, which they validate with ISH (Figure 6H). However, they are strongly encouraged to also validate with ISH putative markers of the subcluster LMCl.2 in Isl2 global or conditional mice. This experiment would significantly strengthen the authors' conclusions (lines 356-366) about which motor pools are affected by Isl2 loss, especially because Pea3 is a marker for multiple motor pools.

We have presented ISH results of six DEGs that showed downregulation in *Isl2* null mice. Remarkably, five of these genes (*C1al3, Etv4, Kcnab1, Prph* and *Anxa2*) were among the Top 30 enriched genes in LMCl.2 (Figure 5h and Supplementary Table S3)*.* In response to the Reviewer’s recommendation, we have additionally included additional ISH results for LMCl.2 markers-*Epha3, Fgf10* and *Sema5a.* Our observations indicated that the transcripts of these markers were exclusively present in Gl motor pools, with their expression levels being diminished in *Isl2* null mice (Figure 6i).

Page 10: **“**For instance, *Epha3*, *Fgf10*, and *Sema5a* transcripts were enriched in Gl motor pools and their expression was reduced in *Isl2* KO, as validated by in situ hybridization analysis (Figure 6i, Supplementary Table S3). In conclusion, our integrated approach, combining bulk and scRNA-seq analyses datasets, has proven valuable in identifying new potential motor pool markers and uncovering novel transcription target genes.”

The study will be strengthened by conducting hindlimb locomotion analysis in Olig2Cre;Isl2 fl/fl mice.

Regrettably, *Isl2* conditional mice experience mortality within a day after birth, precluding the possibility of conducting behavioral studies. A comprehensive presentation of the outcomes and their interpretation concerning the observed disparity in the mortality of these two mouse lines has been detailed. These details can be referenced in in Table S1, Supplementary Videos 1-3, as well as within the Results and Discussion section (pages 10, 16).

The study will be strengthened by conducting hindlimb locomotion analysis in Olig2Cre;Isl2 fl/fl mice.The text can be significantly improved by eliminating typos and improving grammar. The authors should go carefully through the text, sentence by sentence, and make the necessary changes.

We thank to the Reviewers for bringing our attention to the grammatical errors. In order to improve our manuscript, it has undergone a meticulous editorial process by a professional editor.

In the first section of Results, the authors can cite Thaler et al. (2004), as their results are consistent with that study.

We have cited the reference as the Reviewer suggests in p 6, line 132, 135.

Reviewer #2 (Recommendations for the authors):In my opinion, the writing is at times not very clear and makes it difficult to appreciate the findings and their relevance in the context of the literature. The experiments are well executed and appropriately analyzed, there are a few things that can be improved, more to it in the following detailed comments:1) I find the introduction difficult to read, it is more like a list of facts than a coherent narrative to properly frame the work. For example, the bit about neuromesodermal progenitors (lines 66-71), while interesting does not fit with the rest of the story. Also, at times the balance between subjects seems a bit off, more emphasis should be given to what is known about Isl2 and Pea3 mutant mice, which is very relevant to the paper.

In order to enhance the clarity of our findings and accentuate key aspects of our study, we have made the decision to eliminate certain less pertinent sections. Simultaneously, we have enriched the manuscript by incorporating a more comprehensive overview of prior investigations involving mutant mice of *Isl2* and *Pea3.* The details of these enhancements are elaborated below.

Page 3: “More than 50 limb muscles in tetrapods are innervated by distinct limb motor pools, suggesting the involvement of diverse molecular mechanisms in shaping individual limb motor pools during spinal cord development (Sullivan, 1962). Specific LMC motor pools express ETS factors Etv4 and Etv1, playing a role in acquiring motor pool identities. In *Etv4* mutant mice, the position and terminal arborization of specific motor pools were perturbed, resulting in disorganized motor pools and impaired motor control (Livet et al., 2002; Vrieseling and Arber, 2006). GDNF is suggested to act as a peripheral signal derived from limb tissues, guiding major axon bundles toward the hindlimb (Gould et al., 2008; Kramer et al., 2006). Furthermore, GDNF signaling appears to induce Etv4 expression in certain motor pools, as deletion of *gdnf,* its receptor *Ret*, or *Gfra1* results in extinguished *Etv4* expression and affected terminal arborization of motor axons in target muscles (Haase et al., 2002; Livet et al., 2002). However, it remains unclear whether the development of Etv4-expressing motor pools solely depends on GDNF signaling.”

2) Line 150: it is not specified which Isl2 KO mice is used. At the very least there should be a reference (Thaler et al., 2004). However, here it is very important to comment that these are the same mice that have been used in a previous study characterizing the role of Isl2 in spinal motor neuron development. In addition, the mice were originally reported to "die within 24 hr of birth" (Thaler et al., 2004), but the authors found that "approximately 10% survived" (lines 315-317). This important information should be introduced at the beginning when the mouse line is first introduced. It will be immediately relevant as in line 160 adult isl2 KO mice are used.

In order to provide contextual clarity, we have integrated the original reference and added a statement clarifying the utilization of the mutant mouse line established and analyzed by Thaler et al. in our study (Page 6). Comprehensive insights and explanations pertaining to the disparity in mortality between the two mouse lines are thoroughly presented and accessible in Table S1, Supplementary Videos 1-3, and the Results and Discussion section (pages 10, 16).

Page 6: “To explore unknown phenotypes, we conducted further investigations using the mutant mouse line previously generated and analyzed by Thaler et al. (Thaler et al., 2004).”

3) Line 159: "Mislocalized motor neurons remained in the adult Isl2-null lumbar spinal cord with a similar number of motor neurons" Mislocalised motor neurons are not very evident from pictures provided in figure 2h higher magnifications would be helpful. Regarding the "similar number", I could not find quantification of mislocalised motor neurons either in embryos or adults, thus a quantitative statement is not justified. Indeed, quantification would be really good to have at both stages.

We refined the original figures by adjusting their cropping to enhance the visibility of ectopic motor neurons. Additionally, for improved continuity, we re-located figure 2h to figure 7a. To offer a precise quantification of ectopic motor neurons, we have incorporated the cell count results within the text and figures 7b, c.

Page 11: “Overall, the total number of ChAT^+^ motor neurons remained unchanged, and mislocalized motor neurons were observed in the P21 *Isl2-*null lumbar spinal cord (Figure 7a-c).”

4) The statistical difference in MMC and PGC numbers (Figure 2f) observed between WT and KO mice should be briefly commented on, in light of Thaler's results and in the context of the ectopic motor neurons.

We have added some sentences to interpret our quantification results of motor neurons together with Thaler’s results as below.

Page 6: “Our findings align with previous results, including a reduced number of visceral motor neurons, reduced nNOS expression, and abnormal dorsomedial migration of motor neurons, reported by Thaler et al. Our study also reveals the appearance of ectopic motor neurons at the limb levels, primarily consisting of MMC and LMCl cells.”

5) Line 169: "We similarly found that MMC neurons were scattered dorsolaterally and located ectopically, either within the LMC columns or immediately dorsal to the motor columns in Isl2-deficient (Figure 3a)" The markers used in Figure 3a do not allow to clearly distinguish MMC, LMCm and LMCl neurons. A different combination should be used such as Lhx3 (MMC) Isl1 (LMCm) and Hb9 (LMCl). In addition, spatial analysis of columnar and divisional segregation should be provided to better support the statement.

To better distinguish motor columns, we used additional markers Hb9, Lhx3, Lhx1 for our analysis in Figure 3 as the Reviewer suggested. sMN (Hb9), MMC (Hb9, Lhx3), HMC (Hb9, Scip), LMCm (Hb9low, Isl1, Foxp1), LMCl (Hb9high, Lhx1, Foxp1) were carefully defined using these markers. Further, FCU (Scip), CM (Etv4, Isl1), LD (Etv4, Lhx1) motor pools were defined as well. Contour, density and spatial plots of motor columns were provided in Figure 3-supplement 2 and 3.

6) Figure 4b: I believe there must be a mistake as it is clear from the pictures in 4a that there are very few Pea3+ neurons in Isl2 KO mice, however in the plots in 4b (last row the three plots on the right) there are huge red contours for Pea3 neurons more or less similar to control mice. Indeed, the difference is clearly evident in the dot plots in Figure 4 supp 1b.

Initially, we employed an R package to generate contour plots, but this approach neglected the cell density aspect. This resulted in misleading interpretations, suggesting a higher presence of Etv4+cells in *Isl2* KO mice. To rectify this, we have optimized the code to incorporate cell density considerations and subsequently re-created all contour plots in Figure 1, Figure 3—figure supplement 2 and Figure 4—figure supplement 1. To enhance visualization and facilitate accurate comparisons, we have further improved the plots by overlaying the position of cells with Etv4+ cell contours (Figure 4—figure supplement 1).

7) It is great that the authors generated an Isl2 conditional allele, however, its phenotype in combination with olig2::cre is not described other than in the experiments in Figure 5 aimed at studying the downregulation of Pae3. Does the Isl2 cKO mouse recapitulate the overall defects observed in Isl2 KO in terms of motor neuron generation, spatial organisation, connectivity, and behaviour? Comparison of the two can be very informative especially regarding the behavioural and sensorimotor connectivity phenotypes as many other cells (i.e., all somatosensory neurons) express Isl2 during development.

Initially, our experiments primarily involved the use of global *Isl2* KO mice, and more recently, we generated *Isl2* conditional mice to address the issue of cell autonomy. Through meticulous observation, we identified that about 10% of global *Isl2* KO survived, allowing us to effectively examine their postnatal phenotypes. Regrettably, the *Isl2* conditional KO (cKO) mice showed a different survival rate, as they all succumbed within a day after birth. Thus, our ability to perform experiments utilizing postnatal animals was limited.

Detailed results and the interpretation surrounding the observed mortality discrepancy in these two mouse lines have been comprehensively presented in Table S1, Supplementary Videos 1-3, and elaborated within the Results and Discussion section.

Nevertheless, we successfully investigated the progress of NMJ development in *Isl2* cKO at E18.5, just prior to birth, and identified congruent developmental anomalies comparable to *Isl2* null mice. Consequently, the phenotypes we observed during the embryonic stage were identical in both mouse lines, i.e., dispersed motor neurons, Etv4 expression absence in selective lumbar motor pools, disrupted NMJ formation within those pools. The new obtained data, along with pertinent explanations, have been integrated into Figure 8—figure supplement 2 and incorporated within the Results and Discussion section, as outlined below.

Page 12: “Due to the postnatal lethality of *Isl2* cKO, we analyzed the muscles of these mice at E18.5, prior to birth, when NMJ formation is in progress (Figure 8—figure supplement 2). GFP-labeled axons arrived in *Isl2-*deficient Gl muscles; however, their terminal arbors were defasciculated and misrouted. Fluorescence intensity of α-bungarotoxin (BTX) at the NMJs was greatly reduced in *Isl2* cKO. Furthermore, AChR clustering appeared to be incomplete, showing slender and elongated NMJs, or some NMJs were abnormally concentrated in proximity to the primary axon shaft in *Isl2* cKO (Figure 8—figure supplement 2b).”

8) Do you observe arborization, sensorimotor connectivity, and NMJ defects shown for Gl motor neurons also in Rf and Tfl motor pools? It seems like it would be the case but it would be a great addition in order to generalise the role of Isl2-Pea3 transcriptional axis on all pools expressing it.

We sincerely appreciate the valuable feedback offered by the Reviewer. Following the suggestion of our Reviewer, we expanded our study to encompass the developmental examination of other Etv4-expressing motor pools, namely Rf and Tfl, in addition to Gl. Our comprehensive analysis uncovered noteworthy developmental impacts on the Tfl motor pools, while the Rf motor pools remained unaffected. Considering the divergence of Tfl axons from Gl projections, it is plausible that these motor pools share analogous developmental genetic programs. A previous study reported that Etv4 expression is transient in Rf motor pools, lasting until E13.5 (De Marco Garcia and Jessell, 2008). Thus, the postnatal development of Rf motor pools could be independent of Etv4. These results support the notion that Isl2 exerts both a universal and distinct influence on Etv4+ motor pools, albeit to varying extents. We have supplemented our findings with illustrative images and quantification data (Figure 7d, Figure 8h-j and Figure 8—figure supplement 3), providing further elaboration within the Results and Discussion sections of our manuscript.

Page 11: “Similarly, Tfl motor pools, whose axons diverged from one of the major branches of Gl ones, also displayed shorter and lesser dendritic arbors in *Isl2* KO mice (Gould et al., 2008). The position and dendritic arborization of Rf motor pools remained unchanged in *Isl2* KO mice (Figure 7d). Considering that Etv4 expression has been previously reported to be transient until E13.5 in Rf motor pools, it may not play a significant role in the dendritic arborization of Rf motor pools (De Marco Garcia and Jessell, 2008).”

Page 13: “In Tfl muscle, abnormal AChRs were formed along axon branches at P0, and a drastic loss in AChR clusters and polyinnervation were observed at P28 in *Isl2* KO mice (Figure 8h, i and j). However, Rf muscles, whose motor pools transiently express Etv4, showed normal arrangement of NMJs (Figure 8—figure supplement 3) (De Marco Garcia and Jessell, 2008).”

Reviewer #3 (Recommendations for the authors):Specific concerns:1) The behavioral phenotypes seen in the videos appear to be very dramatic, especially in the hindlimbs. However, the authors show that most hindlimb muscles are properly innervated, with the exception of the Gl. For example, the Rf seems to be properly innervated but its weight is dramatically reduced in Figure 9. What is the cause for this? Do the authors believe that the behavior correlates to defects in Gl development and connectivity or are there non-MN contributions to these phenotypes from sensory neurons and interneurons?

Our observations revealed a significant disruption in the development of both Gl and Tfl motor pools, manifesting in altered neuron branching patterns and impaired NMJ formation. Given the substantial role of Gl and Tfl muscles as hip extensors and supporters of hip movements, their compromised function could adequately explain the observed rigidity in hip movement within *Isl2* null mice. Although the development of Rf motor pools was relatively less affected, we postulate that the reduced muscle weight in Rf could be attributed to a broader hindlimb movement reduction.

To delineate whether the observed defects in global *Isl2* KO mice were solely attributed to motor neurons, we conducted further investigations by characterizing motor neurons in *Isl2* cKO, where Isl2 was specifically absent within motor neurons. Intriguingly, the phenotypes we identified in these two mouse lines were strikingly similar, encompassing dispersed motor neurons, the absence of Etv4 expression in selective lumbar motor pools, and disrupted NMJ formation. Further, our investigations extended to the chicken neural tube, where acute knockdown of *Isl2* likewise abolished *Etv4* expression. Detailed information, including the new data, has been incorporated in Figure 8—figure supplement 2 and incorporated them into the Results and Discussion section as below.

Page 12: “Due to the postnatal lethality of *Isl2* cKO, we analyzed the muscles of these mice at E18.5, prior to birth, when NMJ formation is in progress (Figure 8—figure supplement 2). GFP-labeled axons arrived in *Isl2-*deficient Gl muscles; however, their terminal arbors were defasciculated and misrouted. Fluorescence intensity of α-bungarotoxin (BTX) at the NMJs was greatly reduced in *Isl2* cKO. Furthermore, AChR clustering appeared to be incomplete, showing slender and elongated NMJs, or some NMJs were abnormally concentrated in proximity to the primary axon shaft in *Isl2* cKO (Figure 8—figure supplement 2b).”

2) The connection between Isl2/gdnf signaling and Pea3 expression is not experimentally supported. The authors do show a specific downregulation of Pea3 at lumbar levels of the spinal cord but whether this is mediated through gdnf is not established. Is Ret or any other component of this signaling pathway downregulated in the RNA-seq data? The fact that lumbar explants do not respond to gdnf may suggest this but the connection is not explicitly shown.

We sincerely appreciate the valuable feedback provided by the Reviewer. We have integrated a pertinent description within the Discussion section (Pages 15-16) to elucidate the connection between Isl2/gdnf signaling and Pea3 expression. We also have included additional data illustrating Ret and Gfra expression in *Isl2* null mice (Figure 8—figure supplement 1), along with relevant interpretations presented in the Discussion section (Page 16). We readily acknowledge the Reviewer’s concerns regarding the limitations of the explant experiment, which encompassed a mixture of Pea3-expressing and non-expressing LMC motor axons. Consequently, we have made the necessary adjustment by excluding the in vitro data and, instead, dedicated our efforts to an in-depth discussion of the disparities observed between mice lacking gdnf signaling and those with Isl2 mutations in our study.

Page 15: “Additional compelling evidence that Isl2 mainly functions in Etv4^+^ motor pools comes from the striking similarities and differences observed in the mutant phenotypes of Isl2 and the GDNF-Etv4 pathway, requiring a detailed comparison. GDNF has been recognized as a critical factor for the survival and innervation of motor neurons during early motor neuron development (Henderson et al., 1994). Deletion of *GDNF* or its receptor *Ret* or *Gfra1* leads to impaired innervation of the peroneal nerve, resulting in no or disrupted axon innervation in major hindlimb muscles, such as Edl, Atib, psoas major, Gmax, and Tfl, to varying degrees (Bonanomi et al., 2012; Gould et al., 2008; Kramer et al., 2006). In contrast, *Isl2* mutant mice showed normal axon projection of the peroneal nerve and relatively normal axon innervation in most hindlimb muscles, unlike motor axons of mice deficient in *GDNF* or its receptors. Notably, discrepancies between the roles of GDNF and Isl2 also emerged in types of motor pools affected. The removal of *GDNF*, its receptor *Ret* or *Gfra1,* or *Etv4* has been mostly characterized in CM and LD motor pools, the Etv4-expressing motor pools at the forelimb level, but their roles in the lumbar motor pools have not been addressed (Arber et al., 2000; Haase et al., 2002; Livet et al., 2002; Vrieseling and Arber, 2006). In our study, we observed a selective loss of *Etv4* expression only in the lumbar spinal cord of *Isl2* KO mice, specifically in Gl and Tfl motor pools. These motor pools displayed a phenotype almost identical to CM and LD motor pools of *Etv4*-null mice, including mispositioned cell bodies, impaired dendrite and axon arborization, defective NMJs, and reduced sensorimotor connectivity. In addition, acute knockdown of *Isl2* in the chicken neural tube and in motor neuron-specific *Isl2* cKO mice was sufficient to abolish *Etv4* transcripts, suggesting that Isl2 positively regulates *Etv4* expression in a cell-autonomous manner. Thus, GDNF likely plays broader roles in guiding most LMC projections and also promoting differentiation of Etv4-expressing motor pools at both limb levels. In contrast, Isl2 plays a precise role in fine-tuning the position of motor neurons, including MMC and LMCl, and also functions in selected lumbar motor pools at the hindlimb, particularly in establishing sensorimotor connectivity during the late period of development.

Nevertheless, Etv4 expression in the CM motor pool at the brachial level was intact in both *Isl2* KO mice and cKO mice, implying that another complex genetic program works in conjunction with Isl2 to contribute to segment-specific regulation. For instance, multiple *Hox* genes control GDNF signaling, with *Hoxc6* activating *Ret* and *Gfra1* expression in rostral brachial spinal cords, regulating terminal arborization of CM and LD motor pools, while *Hoxc8* activates *Ret* and *Gfra3* expression in caudal brachial spinal cords, controlling limb innervation of distal nerves (Catela et al., 2022). Similar regulatory pathways involving Hox/Mes in *Ret* expression have been suggested at lumbar levels, with misexpression of *Hoxc10* or *Hoxd10* in chicken embryos resulting in *Ret* expression arising in the neural tube (Catela et al., 2022). Our bulk RNA-seq analysis identified *Hoxc11* as one of the DEGs, indicating a potential compromise of proper lumbar identity in the absence of Isl2. However, the expression of *Ret* and *Gfra1* was not changed in *Isl2* KO mice when assessed by bulk RNA-seq analysis and in situ hybridization, and overall hindlimb motor projections remained intact. Although further investigation is necessary to fully elucidate the hierarchy among Hox, GDNF and Isl2, it is evident that each factor plays a crucial role, to varying degrees, in the proper expression of *Etv4* within proximal hindlimb motor pools.”

3) The loss of synapses in Fig7b is interpreted as Isl2 controlling Gl connectivity. Since this is not a conditional allele, the synaptic loss could be due to defects in sensory neurons. The authors also show an ectopic soma (although this is not really well defined). Is there a difference between ectopic and Gl MNs that settle in the correct position? Are the authors suggesting that cell body position influences connectivity in a way that ectopic MNs lose more connections? This is not quantified or discussed.

We express our gratitude for the valuable comments provided by the Reviewer. Regrettably, the postnatal lethality exhibited by *Isl2* cKO mice precluded our ability to explore Gl connectivity in this context. However, the observed NMJ defects in *Isl2* cKO animals substantiate the pivotal role of Isl2 within motor neurons. In alignment with the Reviewer’s guidance, we undertook the categorization of motor neurons into two distinct zones: zone 1 encompassing motor neurons positioned correctly, and zone 2 encompassing ectopic somata. Our findings unveiled a trend wherein ectopic somata tended to exhibit a lower number of sensory boutons per soma, suggesting a clear influence of cell body positioning on connectivity. To provide a comprehensive representation, we have incorporated pertinent data within Figure 7e-k and incorporated them into the Results section as follows.

Page 11: “Next, we analyzed the synaptic density of vGluT1^+^ boutons on motor neuronal somata. Overall, the density of vGluT1 immunoreactivity in zone 1 was higher than in zone 2, which is expected since more motor neurons reside in zone 1 where more sensorimotor synapses are present. Compared to the control mice, the density of vGluT1 immunoreactivity in *Isl2* KO mice did not change, indicating that overall sensorimotor connection is not affected in them (Figure 7h). However, when we analyzed the number of vGluT1^+^ boutons in contact with soma of Rh-Dex^+^ Gl motor neurons, we observed a great reduction in the number of boutons in *Isl2* KO mice (Figure 7i-k). Furthermore, the number of boutons in zone 2, but not in zone 1, was significantly reduced in *Isl2* KO mice (Figure 7k). Thus, ectopically located Gl motor neurons are more likely to receive fewer proprioceptive sensory inputs compared to the motor neurons in a normal position in the absence of *Isl2*. Collectively, our findings suggest that *Isl2* KO mice displayed reduced and misoriented dendrites of Gl motor pools, leading to defective sensorimotor connections.”

4) The way MN numbers are given throughout the manuscript is confusing. The authors quantify MN/section. However, especially for motor pool counts, this results in a lot of variability since at the beginning and end of each motor pool you have 1-2 MNs, while higher numbers are in the middle. The data should be recalculated and presented as average MN numbers/mouse (n=1 mouse and not 1 section). This would reduce variability within the datasets.

We re-analyzed number of motor neurons as an average number of motor neurons per section derived from one embryo of each genotype in Figure 2 to Figure 7.

5). Figure 1: The authors discuss the different spatiotemporal expressions of Isl1 and Isl2 but this is not shown (maybe inferred from RNAseq). Only one time point is shown in the figure (e12.5). It would be nice to show the temporal expression of both TFs at different embryonic time points and also in combination with motor pool markers. For example, the overlap of Isl2 with Pea3 at brachial and lumbar levels is not shown. Showing which motor pools express high and sustained levels of Isl2 in this figure would be a nice setup for the rest of the story.

To refine the temporal depiction of Isl2 and Pea3 expression dynamics, we have enriched Figure 1d and e with contours or additional images. These results illuminate the spatiotemporal distribution of Isl1, Isl2 and Etv4 at E10.5 and/or E12.5. Furthermore, we have provided further details in Result section as below. In brief, our observations delineate the initiation of Isl2 expression in early postmitotic motor neurons, slightly lagging behind Isl1 across all motor neurons. Meanwhile, Etv4 expression remains confined to specific LMC motor pools.

Page 5: “At E10.5, during motor neuron generation, Olig2^+^ pMNs were initially found in the middle part of the ventral spinal cord, co-expressing Lhx3, a marker for pMNs and MMC (Lee, Lee, Ruiz, and Pfaff, 2005; Thaler, Lee, Jurata, Gill, and Pfaff, 2002) (Figure 1d). As these cells migrated, they transiently expressed markers for postmitotic motor neurons Isl1 and Hb9. *Isl2* expression emerged slightly later, found in more lateral regions across all levels analyzed. To analyze the distribution of Isl2 during the motor neuron maturation, we compared the relative position of three populations: Olig2^+^ pMNs, Olig2^+^Isl1^+^ immature motor neurons, and *Isl2*-expressing motor neurons. Contour and density assessments revealed that *Isl2* expression began slightly after Isl1 expression in immature motor neurons and persisted in postmitotic motor neurons along with Isl1 (Figure 1d). At E12.5, when motor columns became distinct, we identified them based on the expression of motor neuronal markers, including Foxp1 (marker for LMC), Isl1 (marker for LMCm and PGC), Lhx3 (marker for MMC), nNOS (marker for PGC), and Etv4 (marker for some LMC motor pools). Motor pools expressing Etv4 were located in the LMC but not at the thoracic level (Figure 1e). Isl2 exhibited broad expression across all motor columns, unlike other LIM-HD transcription factors that showed specific expression patterns. Overall, while *Isl2* is present in differentiated motor neurons, it is relatively enriched in MMC and LMCl populations, where its expression is more prominent compared to Isl1.”

6). Figure 2-4: MN numbers should be recalculated and presented so that n=1 mouse. It is likely that this might change some of the data. For example, in Figure 3a, there is a visible reduction of Scip+ MNs. When the source data for Figure 3c was reanalyzed, there seems to be a 30% reduction of FCU neurons with a P=0.03 (not significant but close, perhaps would be significant if n>3). Is Isl2 highly expressed in Scip+ MNs at brachial levels (see point for Figure 1). Also the MN positional changes should be quantified in these figures.

We re-analyzed number of motor neurons as an average number of motor neurons per section derived from one embryo of each genotype in Figure 2 to Figure 7. We replaced the image of Scip neurons, which provides a more representative depiction of their average distribution in C6 and C8 segments. We also reanalyzed FCU counts as the average number of MNs per embryo, revealing no significant difference. In addition, we provided density plots and boxplots showing distribution of neurons along V-D and M-L axes in Figure 3 and 4.

7). Figure 6: After comparing their RNAseq data to scRNAseq the authors mention that only 50% seem to be MN-specific. Is this technical (not deep enough sequencing in the scRNA-seq datasets) or are the rest of the genes in interneurons? This relates to the comment about other affected populations in the null mouse. In the in situ data, the ventrolateral pools are indicated, however, the changes appear to be broader, affecting the entire lumbar spinal cord. The nNos data should be shown separately since that is in the thoracic spinal cord.

In our investigation, we conducted bulk RNAseq analysis on tissue samples from the ventral part of embryonic spinal cord tissues. Thus, the analyzed transcripts could potentially originate from either motor neurons or other cell types. To pinpoint genes within motor neurons, we referred to the list of genes established to exist in motor neurons. Employing a threshold filter, we considered genes with a minimum raw read count greater than 1 per gene from the scRNA-seq dataset obtained from embryonic motor neurons. Such threshold filters are commonly used in transcriptomic analyses, often with varying read count threshold ranging from 1 to around 10, contingent upon specific research criteria (Corchete et al., 2020; Evrony, Lee, Park, and Walsh, 2016; Namboori et al., 2021). To capture potential DEGs affiliated with motor neurons without losing valuable candidates, we adopted a minimum raw read count > 1 per gene. However, given that our data emanated from the entire ventral spinal cord and did not further refine genes exclusively expressed in motor neurons, some candidate genes showed broader expression. Regardless of the origin of the transcript, whether motor neuron or broadly expressed, we verified that the levels of transcripts were diminished in specific motor pools of interest in *Isl2* mutant mice, in alignment with our DEG analysis. In our revised manuscript, we have spotlighted ISH results from six DEGs and an additional three enriched genes identified from scRNAseq analyses. These genes collectively exhibit relatively specific expression in Gl motor pools, with a noticeable distinction of its transcript in *Isl2* null mice (Figure 6h, i). Lastly, we have removed nNos data since it is already reported earlier in Figure 2.

Page 10: **“**For instance, *Epha3*, *Fgf10* and *Sema5a* transcripts were enriched in Gl motor pools and their expression was reduced in *Isl2* KO, validated by in situ hybridization analysis (Figure 6i, Supplementary Table S3). Collectively, our integrated approach, combining bulk and scRNAseq analyses datasets, proves valuable in identifying new potential motor pool markers and uncovering novel transcription target genes.”

8). Figure 7: In the text, it is mentioned that the Pea3 pools are Gl, Rf, and Tfl and that a retrograde tracer was injected into individual muscles. However, only the Gl data are shown. What about the other pools? Do they have normal dendrites? It seems that Pea3 is not the whole story as Rf has normal NMJ innervation. What about sensory innervation? Is that synapse loss specific to Gl MNs? What about Rf and Tfl? What about non-Pea3+ pools?

We sincerely appreciate the valuable feedback offered by the Reviewer. Following the suggestion of our Reviewer, we expanded our study to encompass the developmental examination of other Etv4-expressing motor pools, namely Rf and Tfl, in addition to Gl. Our comprehensive analysis uncovered noteworthy developmental impacts on the Tfl motor pools, while the Rf motor pools remained unaffected. Considering the divergence of Tfl axons from Gl projections, it is plausible that these motor pools share analogous developmental genetic programs. A previous study reported that Etv4 expression is transient in Rf motor pools, lasting until E13.5 (De Marco Garcia and Jessell, 2008). Thus, the postnatal development of Rf motor pools could be independent of Etv4. These results support the notion that Isl2 exerts both a universal and distinct influence on Etv4+ motor pools, albeit to varying extents. We have supplemented our findings with illustrative images and quantification data (Figure 7d, Figure 8h-j and Figure 8—figure supplement 3), providing further elaboration within the Results and Discussion sections of our manuscript.

Page 11: “Similarly, Tfl motor pools, whose axons diverged from one of the major branches of Gl ones, also displayed shorter and lesser dendritic arbors in *Isl2* KO mice (Gould et al., 2008). The position and dendritic arborization of Rf motor pools remained unchanged in *Isl2* KO mice (Figure 7d). Considering that Etv4 expression has been previously reported to be transient until E13.5 in Rf motor pools, it may not play a significant role in the dendritic arborization of Rf motor pools (De Marco Garcia and Jessell, 2008).”

Page 13: “In Tfl muscle, abnormal AChRs were formed along axon branches at P0, and a drastic loss in AChR clusters and polyinnervation were observed at P28 in *Isl2* KO mice (Figure 8h, i and j). However, Rf muscles, whose motor pools transiently express Etv4, showed normal arrangement of NMJs (Figure 8—figure supplement 3) (De Marco Garcia and Jessell, 2008).”

To discern whether the defects evident in the global *Isl2* KO mice stem exclusively from motor neurons, we undertook a detailed characterization of motor neurons in *Isl2* cKO, where Isl2 was specifically lacking within motor neurons. Notably, the observed phenotypes in both mouse lines was remarkably similar, i.e., dispersed motor neurons, the absence of Etv4 expression in selective lumbar motor pools, disrupted NMJ formation. This congruence in outcomes was further underscored by our observations in the chicken neural tube, where acute knockdown of *Isl2* led to the distinction of *Etv4* expression.

Additionally, we investigated the spindle innervation of sensory Ia to Gl muscles in *Isl2* null mice. The results indicated an unaffected sensory nerve innervation, suggesting that Isl2 deficiency did not influence sensory nerve connections. It is important to note that, given the preliminary nature of this finding and its relevance ot ongoing studies, we have opted to present these results exclusively for the Reviewers at this stage (see Author response image 1). Also we examined dendrite patterning in Ta motor pools as an example of non-Pea3 motor pools. Our findings revealed no significant changes in dendritic arborization, as depicted below (see Author response image 1).

**Author response image 1. sa2fig1:** Assessment of dendritic arbors in Ta motor pools and proprioceptive innervation in Gl muscles. (a) Representative dendritic arbors of Ta motor pools retrogradely labeled with rhodamin-dextran (red). Radial plots show similar dendritic development in *Isl2* KO. (b) Sensory innervation of muscle spindle of Gl muscles is normal in P28 *Isl2* KO, visualized by YFP expression driven by *Thy1::YFP*.

Our primary focus for synaptic bouton quantification was on Gl motor pools, primarily due to their clear display of ectopic motor neurons apart from those in the typical position. This allowed us to compare the number of synaptic boutons between ectopic and correctly positioned motor neurons. While we suspect that Tfl motor pools might also exhibit similar synaptic defects, these were not analyzed due to their smaller size and challenges associated with locating ectopic neurons. Similarly, other motor pools such as Rf or Ta, which showed relatively normal development, were not subjected to synaptic bouton examination, as the likelihood of synaptic defects in these pools was lower.

References

Arber, S., Ladle, D. R., Lin, J. H., Frank, E., and Jessell, T. M. (2000). ETS gene Er81 controls the formation of functional connections between group Ia sensory afferents and motor neurons. *Cell, 101*(5), 485-498. doi:10.1016/s0092-8674(00)80859-4

Bonanomi, D., Chivatakarn, O., Bai, G., Abdesselem, H., Lettieri, K., Marquardt, T.,... Pfaff, S. L. (2012). Ret is a multifunctional coreceptor that integrates diffusible- and contact-axon guidance signals. *Cell, 148*(3), 568-582. doi:10.1016/j.cell.2012.01.024

Catela, C., Chen, Y., Weng, Y., Wen, K., and Kratsios, P. (2022). Control of spinal motor neuron terminal differentiation through sustained Hoxc8 gene activity. *ELife, 11*. doi:10.7554/*eLife*.70766

Corchete, L. A., Rojas, E. A., Alonso-Lopez, D., De Las Rivas, J., Gutierrez, N. C., and Burguillo, F. J. (2020). Systematic comparison and assessment of RNA-seq procedures for gene expression quantitative analysis. *Sci Rep, 10*(1), 19737. doi:10.1038/s41598-020-76881-x

Dasen, J. S., De Camilli, A., Wang, B., Tucker, P. W., and Jessell, T. M. (2008). Hox repertoires for motor neuron diversity and connectivity gated by a single accessory factor, FoxP1. *Cell, 134*(2), 304-316. doi:10.1016/j.cell.2008.06.019

De Marco Garcia, N. V., and Jessell, T. M. (2008). Early motor neuron pool identity and muscle nerve trajectory defined by postmitotic restrictions in Nkx6.1 activity. *Neuron, 57*(2), 217-231. doi:10.1016/j.neuron.2007.11.033

El-Brolosy, M. A., and Stainier, D. Y. R. (2017). Genetic compensation: A phenomenon in search of mechanisms. *PLoS Genet, 13*(7), e1006780. doi:10.1371/journal.pgen.1006780

Evrony, G. D., Lee, E., Park, P. J., and Walsh, C. A. (2016). Resolving rates of mutation in the brain using single-neuron genomics. *ELife, 5*. doi:10.7554/*eLife*.12966

Gergics, P., Brinkmeier, M. L., and Camper, S. A. (2015). Lhx4 deficiency: increased cyclin-dependent kinase inhibitor expression and pituitary hypoplasia. *Mol Endocrinol, 29*(4), 597-612. doi:10.1210/me.2014-1380

Gould, T. W., Yonemura, S., Oppenheim, R. W., Ohmori, S., and Enomoto, H. (2008). The neurotrophic effects of glial cell line-derived neurotrophic factor on spinal motoneurons are restricted to fusimotor subtypes. *J Neurosci, 28*(9), 2131-2146. doi:10.1523/JNEUROSCI.5185-07.2008

Haase, G., Dessaud, E., Garces, A., de Bovis, B., Birling, M., Filippi, P.,... deLapeyriere, O. (2002). GDNF acts through PEA3 to regulate cell body positioning and muscle innervation of specific motor neuron pools. *Neuron, 35*(5), 893-905. doi:10.1016/s0896-6273(02)00864-4

Harrison, K. A., Thaler, J., Pfaff, S. L., Gu, H., and Kehrl, J. H. (1999). Pancreas dorsal lobe agenesis and abnormal islets of Langerhans in Hlxb9-deficient mice. *Nat Genet, 23*(1), 71-75. doi:10.1038/12674

Henderson, C. E., Phillips, H. S., Pollock, R. A., Davies, A. M., Lemeulle, C., Armanini, M.,... et al. (1994). GDNF: a potent survival factor for motoneurons present in peripheral nerve and muscle. *Science, 266*(5187), 1062-1064. doi:10.1126/science.7973664

Kramer, E. R., Knott, L., Su, F., Dessaud, E., Krull, C. E., Helmbacher, F., and Klein, R. (2006). Cooperation between GDNF/Ret and ephrinA/EphA4 signals for motor-axon pathway selection in the limb. *Neuron, 50*(1), 35-47. doi:10.1016/j.neuron.2006.02.020

Lee, S. K., Lee, B., Ruiz, E. C., and Pfaff, S. L. (2005). Olig2 and Ngn2 function in opposition to modulate gene expression in motor neuron progenitor cells. *Genes Dev, 19*(2), 282-294. doi:10.1101/gad.1257105

Livet, J., Sigrist, M., Stroebel, S., De Paola, V., Price, S. R., Henderson, C. E.,... Arber, S. (2002). ETS gene Pea3 controls the central position and terminal arborization of specific motor neuron pools. *Neuron, 35*(5), 877-892. doi:10.1016/s0896-6273(02)00863-2

Namboori, S. C., Thomas, P., Ames, R., Hawkins, S., Garrett, L. O., Willis, C. R. G.,... Bhinge, A. (2021). Single-cell transcriptomics identifies master regulators of neurodegeneration in SOD1 ALS iPSC-derived motor neurons. *Stem Cell Reports, 16*(12), 3020-3035. doi:10.1016/j.stemcr.2021.10.010

Sullivan, G. (1962). Anatomy and embryology of the Wing Musculature of the domestic fowl (gallus). *Australian Journal of Zoology 10*(3), 458-518.

Thaler, J. P., Koo, S. J., Kania, A., Lettieri, K., Andrews, S., Cox, C.,... Pfaff, S. L. (2004). A postmitotic role for Isl-class LIM homeodomain proteins in the assignment of visceral spinal motor neuron identity. *Neuron, 41*(3), 337-350. doi:10.1016/s0896-6273(04)00011-x

Thaler, J. P., Lee, S. K., Jurata, L. W., Gill, G. N., and Pfaff, S. L. (2002). LIM factor Lhx3 contributes to the specification of motor neuron and interneuron identity through cell-type-specific protein-protein interactions. *Cell, 110*(2), 237-249. doi:10.1016/s0092-8674(02)00823-1

Vrieseling, E., and Arber, S. (2006). Target-induced transcriptional control of dendritic patterning and connectivity in motor neurons by the ETS gene Pea3. *Cell, 127*(7), 1439-1452. doi:10.1016/j.cell.2006.10.042